



# Precipitation rather than wind drives the response of East Asian
# forests to tropical cyclones
Yi-Ying Chen[1] and Sebastiaan Luyssaert[2]
[1]Research Center for Environmental Changes, Academia Sinica, Taipei, 11529, Taiwan
[2]Faculty of Science, Vrije Universiteit Amsterdam, Amsterdam, 1081, The Netherlands
*Correspondence to*: Yi-Ying Chen (yiyingchen@gate.sinica.edu.tw)
**Abstract.** Forests disturbance by tropical cyclones is documented by field studies of exceptionally strong cyclones
and satellite-based approaches attributing decreases in leaf area. The biases that come with such approaches may limit
our understanding of the impact of cyclones in general. This study overcomes such biases by starting the analysis from
the observed storm tracks rather than the observed damage. Changes in forest leaf area in East Asia were assessed by
jointly analyzing the cyclone tracks, climate reanalysis, and changes in satellite-based leaf area following the passage
of $145 \pm 42$ cyclones. Sixty days following their passage, $14 \pm 6\%$ of the cyclones resulted in a decrease and $55 \pm 21\%$
showed no change in leaf area compared to nearby forest outside the storm track. For a surprising $31 \pm 6\%$ of the
cyclones, an increase in leaf area was observed. Further analysis revealed that cyclones bringing abundant precipitation
to dry forest soils in summer could relieve water stress within the storm track increasing its leaf area compared to
vegetation outside the storm track. This observation calls for refining the present-day view of cyclones as agents of
destruction toward a more nuanced vision that recognizes that cyclones could have minor or even positive effects on
leaf area and as such on forest growth.
**Main Text**
Each year almost 30 cyclones, about one-third of the world's tropical cyclones, develop over the Pacific Ocean north
of the equator (Landsea, 2000) where a subtropical ridge steers them mainly west and northwest towards Eastern Asia,
where 90 % make landfall. The majority of the tropical cyclones in the northwestern Pacific basin develop between
June and November (Bushnell et al., 2018) and more than half acquire typhoon strength (WMO, 2017). The four most
powerful typhoons in the region since 1999, i.e., Morakot in 2009, Megi in 2010, Haiyan in 2013, and another typhoon
also named Megi in 2016, claimed over 7,000 lives, left 1,700 missing, and destroyed over 10 billion USD worth of
infrastructure and crops according to compilations of mostly local news sources (Yang et al., 2014; Bowen, 2016; Lu
et al., 2017; OCHA, 2010). Although natural ecosystems, such as forests, have adapted to recurring high wind speeds
(Eloy et al., 2017; Louf et al., 2018; Curran et al., 2008), stem breakage is almost unavoidable at wind speeds above
40 m s$^{-1}$ (Virot et al., 2016) but has been widely reported at wind speeds well below this threshold together with other
damage (Tang et al., 2003; Chiu et al., 2018; Chang et al., 2020a). Despite the economic importance of forests in the
region (Barbier, 1993; Vickers et al., 2010), an overall assessment of the damage of tropical cyclones on forest
resources is still lacking.




By jointly analyzing cyclone tracks (JTWC, 2019), climate reanalysis data (ECMWF, 2019), and satellite-based
proxies of soil dryness (Beguería et al., 2014), land cover (ESA, 2017), and leaf area (Martins et al., 2020), we
estimated: (a) the potential forest area damaged by tropical cyclones, (b) the impact of tropical cyclones on leaf area,
and (c) the main drivers of this impact. Previous studies attributed decreases in leaf area or related satellite-based
indices to different disturbance agents including cyclones (Ozdogan et al., 2014; Takao et al., 2014; Honkavaara et
al., 2013; Forzieri et al., 2020). A damage-based approach is designed to identify only decreases in leaf area, thus
failing to identify events in which tropical cyclones left the leaf area unaltered or increased it. In contrast, this study
starts the analysis from the actual storm tracks which allows for an unbiased assessment of the impact of cyclones on
forests (Blanc and Strobl, 2016).

The land area affected was identified for each of the 580 tropical cyclones that occurred in the study region between
1999 and 2018, considering that cyclone-driven damage could only occur within the storm track at locations that
experienced high wind speeds and/or high precipitation. Pixels within the storm track for which the threshold values
were exceeded were classified as affected areas, the remaining pixels served as a cyclone-specific reference area. The
uncertainty derived from defining the width of the storm track (Willoughby and Rahn, 2004) and determining which
wind speeds and amounts of precipitation could result in damage are accounted for by an ensemble of nine related
definitions with different threshold values (**Table A1**). Uncertainties reported in this study represent the standard
deviation across the nine definitions for the affected area.

Since 1999, 224 ± 69 Mha of forest in the study region experienced conditions that may have resulted in cyclone-
driven damage, at least once every decade (**Fig. 1A**). At decadal or longer return intervals, a single cyclone may greatly
affect ecosystem functioning, forest structure and species composition of the forest (Xi, 2015; Castañeda-Moya et al.,
2020). No less than 54 ± 26 Mha, including 70 % of the tropical forest in the region, experienced potentially damaging
conditions at least once per year, and are thus classified as being under chronic wind stress (**Fig. 1A**). Lower estimates
from the rain-only definitions closely matched the 70 Mha yr$^{-1}$ that was reported following a similar approach in which
the affected area was defined as a 100 km buffer zone along the storm track (Lin et al., 2020).

Irrespective of the definition of the affected area, the coefficient of variation of the between-year variation in
potentially damaged areas ranged from 15 to 20 % (**Fig. 1B**). Excluding the four most powerful typhoons that occurred
in the region since 1999 changed the average coefficient of variation from 17 to 16 %. This suggests that the most
powerful typhoons make only a small contribution to the total annually potentially affected area in the region. A recent
literature review reported, however, that 66 % of the research papers in this area have examined the effects of only
about 6 % of the most powerful cyclones (Lin et al., 2020). The relatively small contribution of those events to the
potentially damage area suggests that in regions with frequent tropical storms, disturbance ecology would benefit from



broadening its scope by examining the effects and recovery of a representative sample of tropical cyclones, rather than
focusing on the most devastating events.

The different definitions of affected area (**Table A1**) consistently show a high potential for forest damage over island
and coastal regions located between 10- and 35-degrees latitude (**Fig. 1C**). Although damage potential is the outcome
of an interplay between cyclone frequency, cyclone intensity and the presence of forests, the high potential in this
region is largely driven by the frequency of tropical cyclones (**Fig. A1**), i.e., two or more cyclones making landfall
per year. Depending on how the affected area is defined, there is a second region located between 40 and 50 degrees
north with a high potential for storm damage (**Fig. 1C**). In this region, the potential damage is the outcome of the high
forest cover resulting in a strong dependency on the assumed width of the storm track (**Fig. A1**).

The impact of a tropical cyclone on leaf area was calculated based on the adjusted Hedge's effect size by comparing
the change in leaf area before and after the cyclone in the affected area with the change before and after the cyclone
in the reference area for each individual cyclone (**Eq. 1**). Using a reference area that is specific to each cyclone means
that seasonal dynamics related to leaf phenology and seasonal monsoons can be accounted for in the effect size, which
is a unitless description of the mean change in leaf area normalized by its standard deviation (**Eq. 1**). A positive or
negative effect size respectively denotes an increase or decrease in leaf area following the passage of a tropical cyclone.

A total of $316 \pm 22$ tropical cyclones or $54 \pm 4$ % of the storm events under study could not be further analysed (**Table**
**A1**) because leaf area index (LAI) observations were missing from either the affected area, the reference area, or both,
thus violating the requirements for calculating the effect size (**Eq. 1**). Of the remaining $264 \pm 22$ tropical cyclones,
only $145 \pm 42$ passed the additional quality checks necessary to be retained for further analysis in this study: (i) have
a less than $0.5$ $m^2$ $m^{-2}$ difference in the leaf area between the reference and affected area prior to the passage of the
storm signifying that prior to the storm the reference area is indeed similar to what will become the affected area; and
(ii) have an effect size that is larger than the noise of the remotely sensed leaf area. Despite the loss of around 75 %
of the events, the quality control criteria resulted in an unbiased sample in terms of wind speed (**Fig. A2**).

The effect size of $79 \pm 31$ events was less than the noise of the remotely sensed change in leaf area suggesting that for
$55 \pm 21$ % of the cyclones, the change in leaf area 60 days after a cyclone passed was too small to distinguish it from
the noise of present-day remote sensing technology. Nevertheless, ecological theory predicts forest dwarfing in regions
with high cyclone frequencies directly through gradual removal of taller trees over many generations (Lin et al., 2020;
McDowell et al., 2020) and indirectly through the loss of nutrients (Tang et al., 2003; Lin et al., 2011). Where forest
dwarfing has occurred, it might be hard to observe the short-term effects of an individual tropical cyclone on forest
structure and function (Mabry et al., 1998). Following the terminology of this study, a neutral effect size over regions
with high return frequencies would be consistent with structural adaptation to frequent cyclones. Indeed, for regions
that experience over 4.5 cyclones per year, the mean effect size was almost zero (**Fig. A3**).




Tropical cyclones have been widely observed to defoliate and disturb forests because of limb breaking, uprooting,
stem breakage and landslides following high wind speeds and heavy precipitation (Wang et al., 2013; Uriarte et al.,
2019; Chambers et al., 2007; Douglas, 1999; Lin et al., 2011). Nevertheless, in this study, only $14 \pm 6$ % of the
observed cyclones resulted in a detectable reduction in leaf area as a direct effect of limb breakage, uprooting, stem
breakage and landslides, 60 days after their passage. On the other hand, for $31 \pm 6$ % of the cyclones an increase in
leaf area was observed, leading to the question: which conditions lead to an increase (or a reduced decrease) in leaf
area between the affected and control areas 60 days following the passage of a tropical cyclone?

To answer this question, two groups of meta-data were compiled for each of the $145 \pm 42$ tropical cyclones that passed
the quality checks, the first group consisting of five characteristics describing the land surface before the passage of a
cyclone and the second group containing five characteristics of the cyclone itself (**Table A2**). Following factorial
analysis to identify collinearity between the meta-data in the same group, the explanatory power of the meta-data was
quantified as a decrease in the accuracy of a random forest analysis (**Fig. 2**). The random forest analysis was repeated
12 times with different combinations of largely uncorrelated meta-data (**Table A3**). Each random forest analysis
included the effect sizes and meta-data for all nine definitions of affected area to account for this specific source of
uncertainty.

The statistical analysis showed that accumulation of precipitation during the passage of a cyclone over land makes the
largest contribution to the accuracy of the random forest analysis. Randomizing this variable decreased the accuracy
of the random forest analysis by 20 to 26 % (**Fig. 2**). Soil dryness quantified as the standardized precipitation and
evapotranspiration index (SPEI) at the time of landfall was the second most important variable contributing 2 to 17 %
whereas the other meta-data contributed relatively little (-4 to 7 %) to the accuracy of the random forest analysis.
Subsequently, the six meta-data with the highest explanatory power were used to build a single regression tree to
obtain the environmental drivers and their cut-off values that would best explain the change in leaf area following the
passage of a tropical cyclone (**Fig. 3**). In the remainder of this report we focus on the unexpected result, i.e., the
increase in leaf area following the passage of a tropical cyclone.

Cyclones bringing abundant precipitation ($\geq$ 19 mm) during summer months (i.e., after month 6.5) when the forest
soil was dry (SPEI $\leq$ -0.74) resulted dominantly (60 to 70 %) in an increase in leaf area along the storm track (**Fig. 3**).
The vegetation response was thought to be the outcome of two elements: (a) cyclones making landfall in June, July
and August bring 30 to 50 % of the annual precipitation in coastal areas in the study domain (**Fig. A4**) and are thus
substantial sources of precipitation. The importance of the precipitation brought by tropical cyclones is confirmed by
domain-wide changes in the Standardized Precipitation-Evapotranspiration Index showing that 1070 of the 1309 (82
%) cyclones increased soil wetness, and (b) given that much of the study domain has a monsoon climate with relatively
little rain in the fall and winter months, the implication is that summer droughts might, for evergreen vegetation, have



lasting effects until the next growing season (Chou et al., 2009) unless the drought was ended before the dry season
begins. Cyclones, especially those later in summer could bring the precipitation to end summer droughts. For the mid-
latitudes, including Korea, China, Taiwan, and Japan, dry summers see an increase in the number of tropical cyclones
making landfall which often end the summer drought (Yoo et al., 2015). In South Korea, for example, at least 43 %
but possibly as much as 90 % of the summer droughts in coastal regions were abruptly ended by a tropical cyclone
(Yoo et al., 2015). Based on our analysis of the Standardized Precipitation-Evapotranspiration Index, 214 of the 1309
(16 %) tropical cyclones in East Asia ended a drought.

An increase in leaf area, following the passage of a tropical cyclone, thus requires three conditions to co-occur: (a) a
dry spell, (b) a cyclone making landfall in the region experiencing the dry spell, and (c) the cyclone bringing abundant
precipitation to mitigate the soil dryness. Meeting all three conditions at the same time seems unlikely unless there is
a physical relationship between summer droughts (a) and tropical cyclones (b). During dry years, a meridional dipole
system has been observed in the mid-latitude regions of East Asia with a high pressure system in the region of 40-50
N and 150-160E where it is causing the dry spell, and the low pressure system in the region of 20-30N and 120-150N.
When such a dipole exists, tropical cyclones generated from the monsoon trough over the West Pacific Ocean are
steered through the trough in between the high- and low-pressure systems towards and then along the coast of East
Asia (Choi et al., 2010). While travelling along the edges of the high pressure system, the tropical cyclone may disturb
the circulation, resulting in an unfavourable environment to sustain the dipole (Choi et al., 2011; Kubota et al., 2016)
and bringing precipitation to the dry region that was under the high pressure system.

By studying a representative sample of tropical cyclones (in terms of storm intensity) (**Fig. A2**), we have shown that
over half of the tropical cyclones, i.e., $55 \pm 21$ %, caused little to no damage to forest leaf area, suggesting that forest
dwarfing is a general structural adaption in the study region. Moreover, a third, i.e., $31 \pm 6$ % of the cyclones in East
Asia resulted in an increase in forest growth, because these storms relieved water stress within their track or even
ended summer droughts. The observed frequency of positive vegetation responses to cyclones suggests that the present
day vision of cyclones as agents of destruction (Altman et al., 2018; Negrón-Juárez et al., 2010; Nelson et al., 1994)
should be refined toward a recognition that, depending on the environmental conditions prior to the storm and the
characteristics of the storm itself, cyclones could also have limited destructive effects (Lin et al., 2020) or even positive
effects on forest growth (Castañeda-Moya et al., 2020; Chang et al., 2020b). As both cyclones (Mei and Xie, 2016)
and droughts (Zhao and Dai, 2017) are expected to continue to intensify with global warming, the net direct effect
through relieved water stress and indirect effect through possible connections with fire activities (Stuivenvolt Allen et
al., 2021) remains highly uncertain**.**

**Materials and Methods**
**Cyclone track and track diameter**



Since 1945, tropical cyclones in the Western North Pacific Ocean have been tracked and their intensity recorded by
the Joint Typhoon Warning Center (JTWC). The track data shared by the JTWC consist of quality-controlled six-
hourly geolocation observations of the center of the storm with the diameter of the storm being a proxy for its intensity
(JTWC, 2019). For the period under consideration, from 1999 to 2018, the geolocations and diameters are the output
of the Dvorak model (Dvorak, 1984; Dvorak et al., 1990) derived from visible and infrared satellite imagery. Storm
diameters are available starting from January 2003. Prior to this date a generic diameter of 100 km (Lin et al., 2020)
is used in this study. Linear interpolation of the six-hourly track data resulted in hourly track data to fill in any gaps
in the mapping of the cyclone track.

In this study, we focus on East Asia which, given the absence of natural boundaries, is defined as the land contained
within the northwestern Pacific basin that, according to the JTWC stretches from 100 to 150 degrees east and 0 to 60
degrees north. The JTWC compiled track and intensity data for 580 tropical cyclones between 1999 and 2018 in the
northwestern Pacific basin. A shorter time series (1999 to 2018) than the entire length of time available (1945 to 2018)
was analyzed due to the more limited availability of the leaf area index (LAI) data which had to be jointly analyzed
with the track and intensity data to quantify the impact of cyclones on natural ecosystems.

**Area affected by individual cyclones**
The land area thought to be affected by a specific cyclone as well as the reference area for each of the 580 cyclones
that occurred in the study area between 1999 and 2018 were identified based on nine different but related definitions
(**Table A1**). Each definition comprises a combination of at least two out of three criteria, e.g., the diameter of the
cyclone, the maximum wind speed at each location during the passage of the cyclone and accumulated precipitation
at each location during the passage of the cyclone. Each forested pixel within each individual storm track was classified
as either affected area or reference area based on these nine definitions. Differences in the results coming from
differences in the definitions were used throughout the analysis to estimate semantic uncertainties. Uncertainties
related to the estimated diameter of the cyclone, wind speed and precipitation data were not accounted for in the
calculation of the affected and reference areas because they were thought to be smaller than the uncertainty coming
from differences in the definitions themselves.

The underlying assumption behind the definitions is that forests can only be affected by a specific cyclone if they are
located along its storm track. The minimum width of each storm track is the diameter of the cyclone as reported by
the JTWC. Following the observation that over the ocean, the actual wind speed exceeds the critical wind speed for
stem breakage or uprooting (i.e., 17 m s$^{-1}$ ref. Chen et al., 2018) over a distance of at least three times the diameter of
the cyclone (Willoughby and Rahn, 2004), the minimum width of a storm track in which cyclone-related forest damage
could occur is defined as three times the diameter recorded by the JTWC although wind speeds drop dramatically
when cyclones make land fall (Kaplan and Demaria, 2001). The minimum width of a storm track over land should,



therefore, be reduced compared to the observations over the ocean. This study used three different widths to define a
storm track, i.e., two, three or four times the recorded diameter (**Table A1**).

Being located within the track of a specific cyclone is essential but not sufficient for damage to occur. Within a storm
track, only forested pixels that experienced high wind speeds or high precipitation were counted as in the potentially
affected area. Forest pixels that were located within the storm track but did not experience high wind speeds or high
precipitation were counted as in the reference area. Note that to better account for the uncertainties arising from this
approach, the threshold values for wind speed and precipitation were also increased as the track diameter increased
(**Table A1**). For a narrow storm track it is reasonable to assume that there would be damage shown in all pixels except
those where wind speed or precipitation did not exceed a relatively low threshold value. For wide storm tracks the
opposite applies; it is reasonable to assume that few of the pixels would show damage except where wind speed or
precipitation exceeded relatively high threshold values.

Data sources for the geolocation and diameter of an individual cyclone are described in detail in 'Cyclone track and
diameter'. Wind speed and precipitation data were extracted from the ERA5-Land reanalysis data for land (ECMWF,
2019). The ERA5-Land reanalysis dataset has a spatial resolution of 9 km x 9 km and a time step of 1 hour. It is the
product of a data assimilation study conducted with the H-TESSEL scheme by ERA5 IFS Cy45r1 and nudged by
climatological observations (ECMWF, 2018). The Cy45r1 reanalysis dataset shows statistically neutral results for the
position error of individual cyclones (ECMWF Confluence Wiki: Implementation of IFS cycle 45r1). The spatial
representation of the reanalysis data is reported to compare favorably with observational data (Chen et al., 2021)
outside the domain of this study. No reports on similar tests for the current study domain, i.e., East Asia, were found.
Furthermore, land cover maps released through the European Space Agency's (ESA's) Climate Change Initiative
(ESA, 2017) were used to restrict the analysis to forests. The CCI maps integrate observations from several space-
borne sensors, including MERIS, SPOT-VGT, AVHRR, and PROBA-V, into a continuous map with a 300 m
resolution from 1994 onwards.

Wind speed and precipitation data were spatially disaggregated and temporally aggregated to match the spatial and
temporal resolution of the ESA leaf area index (LAI) product (see below). Maximum wind speed and accumulative
precipitation were aggregated over time steps to match the 10-day resolution of the ESA LAI product. We preserved
the temporal resolution of the land cover map but aggregated the spatial resolution from 300 m to 1 km to match the
resolution of the ESA LAI product. During aggregation, the majority of land cover at the 300 m resolution was
assigned to the 1 km pixel resolution.

The oceanic Nino index (ONI) was retrieved from NOAA (NINO SST INDICES (NINO 1+2, 3, 3.4, 4; ONI AND
TNI), 2019). The oceanic Nino index was calculated and defined by comparing the 3-month running mean sea surface
temperature over the region from 5 degrees north to 5 degrees south and from 170 degrees west to 120 degrees west



with the 30 year climatology of sea surface temperature over the same region (Trenberth andStepaniak, 2001; The
climate data guide: Nino SST indices (Nino 1+2, 3, 3.4, 4; ONI and TNI)). A monthly seasonal oceanic Nino index
was used in this study. According to this method, El Nino events are characterized by an oceanic Nino index exceeding
0.5 K and La Nina events by an oceanic Nino index below -0.5 K. These thresholds relate to a warmer or a cooler
ocean state in the central tropical Pacific.

**Impact on leaf area of an individual cyclone**
Version 2 of ESA's Climate Change Initiative product was used to calculate leaf area (LAI) in this study. The product
has a 1 km spatial resolution, a 10-day temporal resolution, and is available from 1999 onwards. The default LAI
product is distributed as a composite image using at least six valid observations on a pixel within a 30-day moving
window (Verger et al., 2014). The composite image is drawn from satellite-based observations of the surface
reflectance in the red, near-infrared, and shortwave infrared from SPOT-VGT (from 1999 to May 2014) and PROBA-
V (from June 2014 to present). Gaps in missing observations are filled by the application of a relationship between
local weather and LAI dynamics. Gap filling resulted in errors on the LAI estimates of less than 0.18 (ref. (Martins et
al., 2020)). The spatiotemporal resolution of the LAI products was the coarsest of all data products used and therefore
determined the spatiotemporal resolution of the analysis as a whole. Moreover, the availability of the LAI product
determined the starting date for the study.

The impact of cyclones on leaf area was calculated by comparing the change in leaf area before and after the cyclone
in the affected area with changes before and after the cyclone in the reference area for each individual cyclone. In this
approach, the reference area serves as the control for the affected area, given that reference area and the affected area
may have a different size, the adjusted Hedge's effect size (Rustad et al., 2001) can be used to calculate the effect size
of an individual cyclone on leaf area (**Eq. 1**). Using a reference area that is specific to each cyclone's seasonal
dynamics, such as leaf phenology, is accounted for in the effect size. Effect size is thus a unitless quantifier which
describes the mean change in state, obtained by normalizing the mean difference in leaf area with the standard
deviation (**Eq. 1**). A positive or negative *ES* value indicates, respectively, an increase or decrease in leaf area following
the passage of a cyclone:

$$ES = \frac{(\overline{LAI}_{bef} - \overline{LAI}_{aft})_{aff} - (\overline{LAI}_{bef} - \overline{LAI}_{aft})_{ref}}{\sigma},$$  [1]

where *ES* is the event-based effect size for leaf area. The upper bar represents the mean of LAI in either the reference
(*ref*) or the affected (*aff*) area. The subscripts *ref* and *aft* denote the observation dates before and after the cyclone; $\sigma$
denotes the standard deviation of all observations within the storm track. Given the 10-day frequency of the ESA LAI
product, two LAI maps are used for the calculation of the *ES*, one to characterize the LAI 1 to 10 days before the
cyclone and the other to characterize the LAI 60 to 70 days after the cyclone. To distinguish between the affected and



reference areas the effect sizes were calculated for each event using the nine definitions. After applying the quality
control criteria (see below) a different number of events was available for each definition (**Table A1**).

The 60-day time frame was a compromise to avoid excessive data gaps in the LAI product when using a composite
LAI product. Because the LAI product reports LAI values within a 60-day window, the analysis had to be refined so
that this 60-day window never included the cyclone. The offset between the cyclone and a LAI observation from the
composite ESA LAI product was calculated by subtracting the date of the cyclone from the last observation date of
the LAI composite data before the cyclone or first observation date of the LAI composite data after the cyclone. Pixels
with a negative offset indicated that the composite data were likely to include observations from both before and after
the cyclone and were therefore discarded in the calculations of the effect size.

Starting the analysis from the actual storm tracks, as was the case in this study, allows for an unbiased assessment of
the impact of cyclones on forests (Blanc and Strobl, 2016), in contrast to studies that attribute decreases in leaf area
or related satellite-based indices to different disturbance agents including cyclones (Ozdogan et al., 2014; Takao et al.,
2014; Honkavaara et al., 2013; Forzieri et al., 2020). By design, the latter approach is not capable of identifying neutral
or positive impacts of cyclones on leaf area.

**Quality control**
The calculation of the effect size relies on having a similar LAI between the area that will become the affected area
and the area that will become the reference area after the passage of a cyclone. If the difference in LAI between the
reference and the affected area was over -0.25 but less than 0.25, the effect size calculated for this event was included
in subsequent analyses. The 0.25 threshold was derived through error propagation by considering that "similar LAI"
implies that the difference in LAI between the reference and affected area should be zero before the event. The
uncertainty from gap-filling satellite-based LAI products, i.e., 0.18 (ref. (Martins et al., 2020)) was used to derive a
reasonable threshold. Given that each LAI measurement may come with an uncertainty of 0.18 the difference between
two such measurements comes with an uncertainty of 0.25 ($\sqrt{0.18^2 + 0.18^2}$).

The uncertainty of ES calculation through error propagation in equation (**Eq. 1**) is:

$$\delta ES = |ES| * \sqrt{\left(\frac{\delta X}{X}\right)^2 + \left(\frac{\delta Y}{Y}\right)^2},$$     [2]

where X is the nominator and Y is the denominator of **Eq. 1**. Given that each LAI observation is assumed to have an
uncertainty of 0.18, $\delta X$ is constant at 0.36. The $\delta Y$ can be calculated by $\sqrt{n * (0.18)^2}/n$, where n is the number of
available observations. For each event, the quality of the ES calculation was examined by comparing the actual *ES* to





its uncertainty $\delta ES$. Events for which $ES < \delta ES$ were not further analyzed. Events with an effect sizes between -0.18
and 0.18 were classified as neutral.

**Multivariate analysis**
Each tropical cyclone was characterized by its: (1) latitude of landfall (degrees); (2) intensity of the tropical cyclone
(m s$^{-1}$); (3) month of landfall; (4) maximum wind speed during passage over land (m s$^{-1}$); (5) accumulated rainfall
during passage over land (mm); (6) accumulated rainfall on land 30 days prior to landfall of the cyclone (mm); (7)
affected area during passage over land (Mha); (8) leaf area 30 days prior to landfall (m$^2$ m$^{-2}$); (9) Standardized
Precipitation Evapotranspiration Index (SPEI) (mm mm$^{-1}$) as a drought proxy; and (10) oceanic Nino index the month
of landfall (K).

Characteristics 1 to 4 were retrieved from the JTWC database as detailed in 'Cyclone track and track diameter'.
Characteristics 5 to 6 were retrieved from the ERA5-Land reanalysis data for land (ECMWF, 2019) and characteristic
7 from the analysis combining cyclone track, cyclone diameter and ERA5-Land reanalysis, as explained in 'Area
affected by individual cyclones'. Characteristic 8 was taken from the LAI analysis as explained in 'Impact on leaf area
of an individual cyclone'. For characteristic 9, Standardized Precipitation Evapotranspiration Index with a half-degree
by half-degree spatial resolution and a 10-day temporal resolution was used and combined with the cyclone masks
created in 'Area affected by the individual cyclone'. Characteristic 10, the oceanic Nino index, was retrieved from
NOAA (NINO SST INDICES (NINO 1+2, 3, 3.4, 4; ONI AND TNI), 2019).

The characteristics were separated into two groups describing the condition of the land and ocean prior to the event
("prior conditions" or PC group) and the characteristics of the tropical cyclone itself ("tropical cyclone characteristic"
or TCC group). The prior conditions group contained: pre-event LAI, pre-event drought state, pre-event accumulative
rainfall, oceanic Nino index, and month. Characteristics such as maximum wind speed, accumulative rainfall, cyclone
intensity, affected area, and latitude were used to describe the cyclone itself.

Factor analysis (Revelle, 2017) was used to reveal the collinearity among the selected variables in the "prior conditions"
and "tropical cyclone characteristic" group (**Table A2**). Collinearity was used to create 12 sets of mostly independent
characteristics (**Table A3**) which were used as the input for a random forest tree to identify the characteristics that
best explained the effect size for LAI. The random forest analysis was repeated for each of the 12 sets, but limited to
four-layer random forest trees, to identify the importance of the environmental variables on the tropical cyclone effect
size (not shown). Finally, to reduce the collinearity of the input variables, only the six variables with the highest
explanatory power were used to create a single decision tree which is shown in **Fig. 3.** For this, the recursive
partitioning approach was used with a maximum of five levels and a minimum of 20 samples in each node provided
by the R-rpart package (Therneau et al., 2019).





**Drought analysis**
The Standardized Precipitation Evapotranspiration Index (SPEI), is a proxy index for drought that represents the
climatic water balance and was used to assess the drought of a forest soil before and after the passage of an individual
tropical cyclone. The Standardized Precipitation Evapotranspiration Index data used in this study were retrieved from
the Global SPEI database (SPEIbase v2.6 (Beguería et al., 2014)), which is based on the CRU TS v.4.03 dataset (Harris
et al., 2020). In this study, the temporal resolution of the data was preserved but the spatial resolution was regridded
from the original half-degree to 1 km to match the resolution of the ESA LAI product. The contribution of an individual
tropical cyclone to ending a drought was evaluated by comparing the SPEI from affected and reference areas through
the following equation:

$\delta SPEI = (SPEI_{imon})_{aff} - (SPEI_{imon})_{ref},$                [3]

where $\delta$SPEI is the event-based change in drought. A positive or negative $\delta$SPEI respectively denotes an increase or
decrease in available water resources following the passage of a tropical cyclone. The subscription *imon* represents
the integration time of available water resources in the calculation of the SPEI either in the reference (*ref*) or the
affected (*aff*) area which are defined in previous section. The same time window, i.e., 60-days, was applied for the
calculation of $\delta$SPEI and event-based effect size for LAI.

**Acknowledgments**
Y.Y.C. would like to thank the National Center for High-performance Computing (NCHC) for sharing its
computational resources and data storage facilities. Y.Y.C. was funded through the Ministry of Science and
Technology (grant MOST 109-2111-M-001-011 and grant MOST 110-2111-M-001 -011).

**Data availability**
R-Scripts and all input data to performing the analysis and creating the plots can be found in the following web-based
repository https://doi.org/10.5281/zenodo.6459795. The database of event-based effect sizes, surface properties and
cyclone properties for each of the 1309 events (i.e., 145 ±42 unique tropical cyclones analyzed for nine related
definitions) can be accessed at:
https://myspace.sinica.edu.tw/public.php?service=files&t=_e2vJFnlASIdGgtvnfcqcXAa51-
aTChcjUgAJXk2mHjoZ1thVek8W9yeJx13GeHb

**Author Contributions**
Y.Y.C. and S.L. designed the study. Y.Y.C. investigated and visualized the results. Y.Y.C. and S.L. contributed to the
interpretation of the results. S.L. wrote the original draft. S.L. and Y.Y.C. reviewed and edited the manuscript.

**Competing Interest Statement:** The authors declare no competing interests.



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

**Figures and Tables**

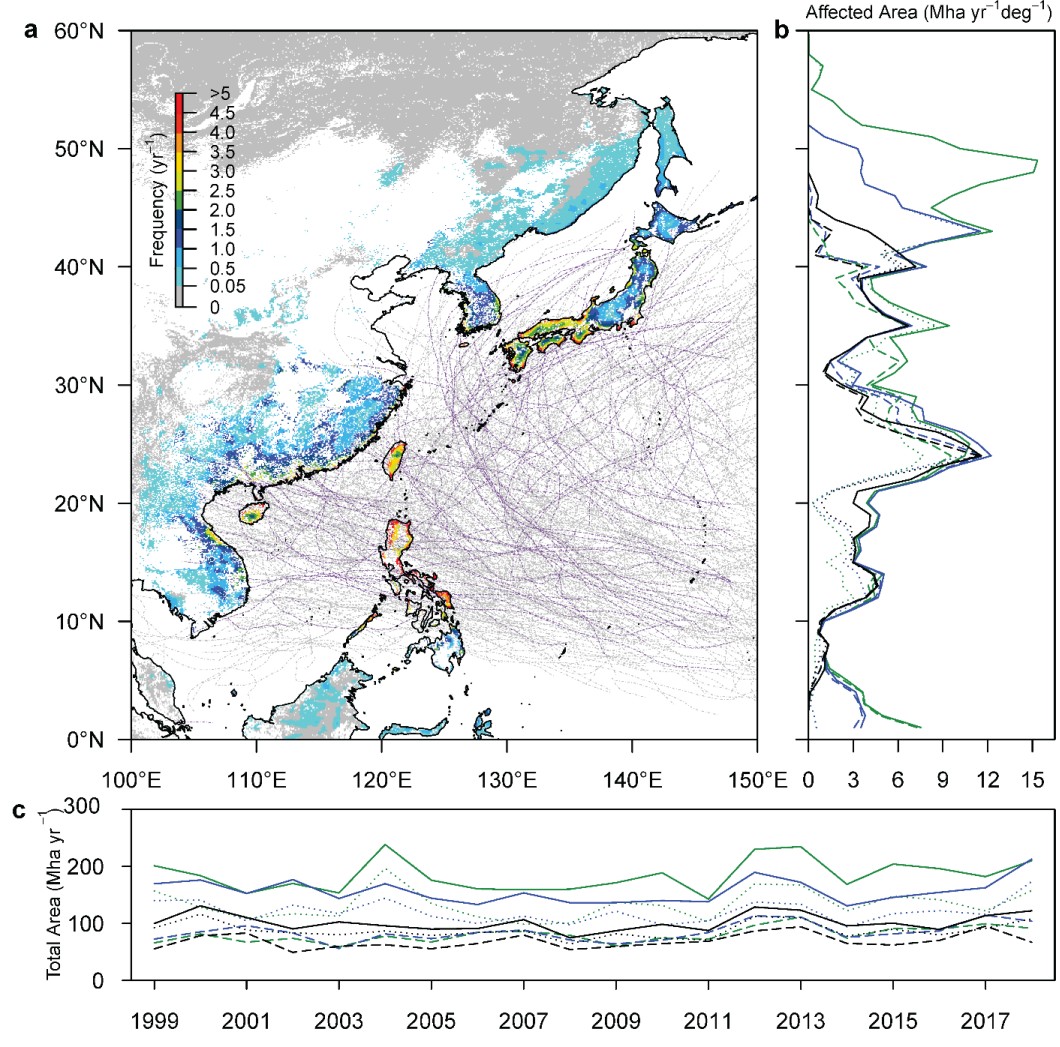


**Figure 1.** Spatial and temporal patterns of potential forest damage by tropical cyclones in East Asia. (**A**) Return
frequency (yr⁻¹) of tropical cyclones between 1999 and 2018. Pixels where forest is the main land cover are shaded.
The color of the shading represents the return frequency of tropical cyclones based on definition 3b for the affected
area (Table A1). The dot-dashed lines show the cyclone tracks between 1999 and 2018. The purple lines indicate the
cases passed the QC/QA criteria used in this study. (**B**) Temporal dynamics of the total potentially damaged forest
area (Mha yr⁻¹) for all nine definitions of affected area. (**C**) Latitudinal gradients of potentially damaged forest area
(Mha yr⁻¹ deg⁻¹) between 1999 to 2018 for all nine definitions of affected area. The dotted lines show the "wind



only" definitions (group 1), the dashed lines show the "rainfall only" definitions (group 2), and the solid lines show
the "combined" definitions (group 3). The black, blue and green colored lines represent definitions a, b and c,
respectively, within each group. Definitions are detailed in Table A1.





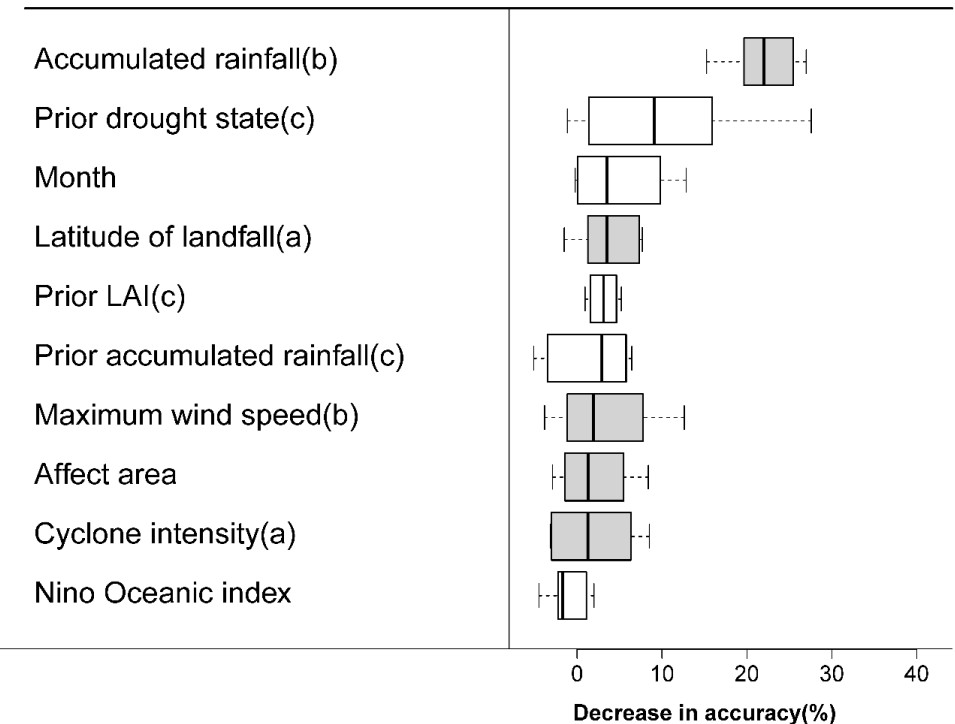


**Figure 2.** Importance of the five surface (white) and five cyclone (grey) characteristics in explaining the LAI response
to the passage of a tropical cyclone. The boxplots show the 95, 75, 50, 25 and 5 percentiles of the decrease in accuracy.
The letters a, b and c following the label of a characteristic indicate collinearity between the variables (Table A2).
Each boxplot contains the results of 12 random forest analyses fitted with different combinations of largely
uncorrelated characteristics (Table A3). Each random forest analysis is based on 1309 cases coming from the 145 ±
42 individual tropical cyclones for which the impact was quantified according to nine related definitions (Table A1).
The medians were used to sort the cyclone and surface characteristics according to decreasing importance.



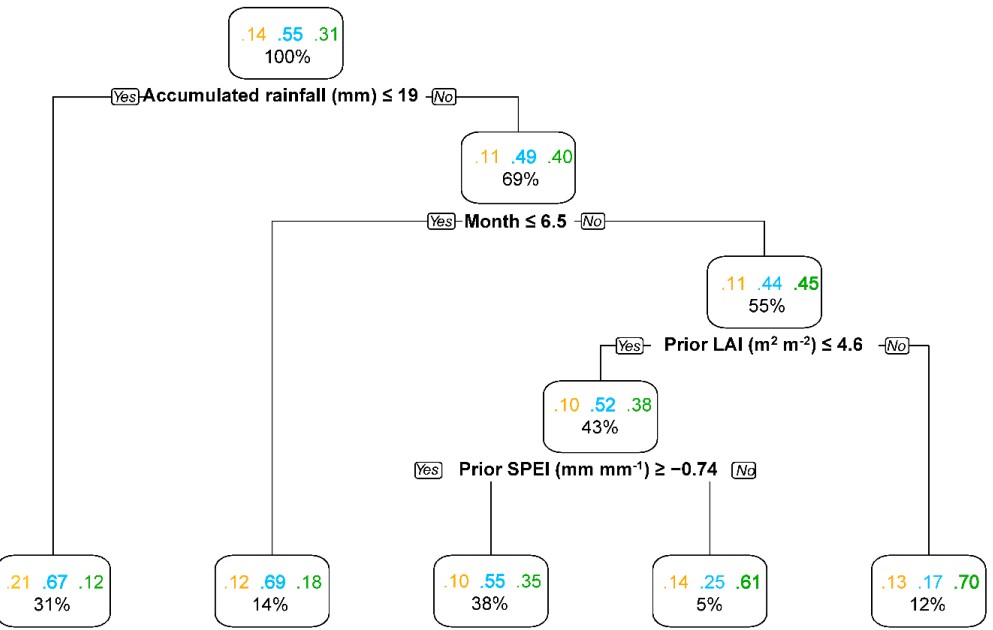


**Figure 3.** Environmental drivers contributing to an increase of LAI following the passage of a tropical cyclone. The

fractions of a negative, neutral and positive effect size are listed for each box in respectively orange, blue, and green.

The number of events is listed as the percentage of the total number of events in the random tree (n=1309).


**Appendix**

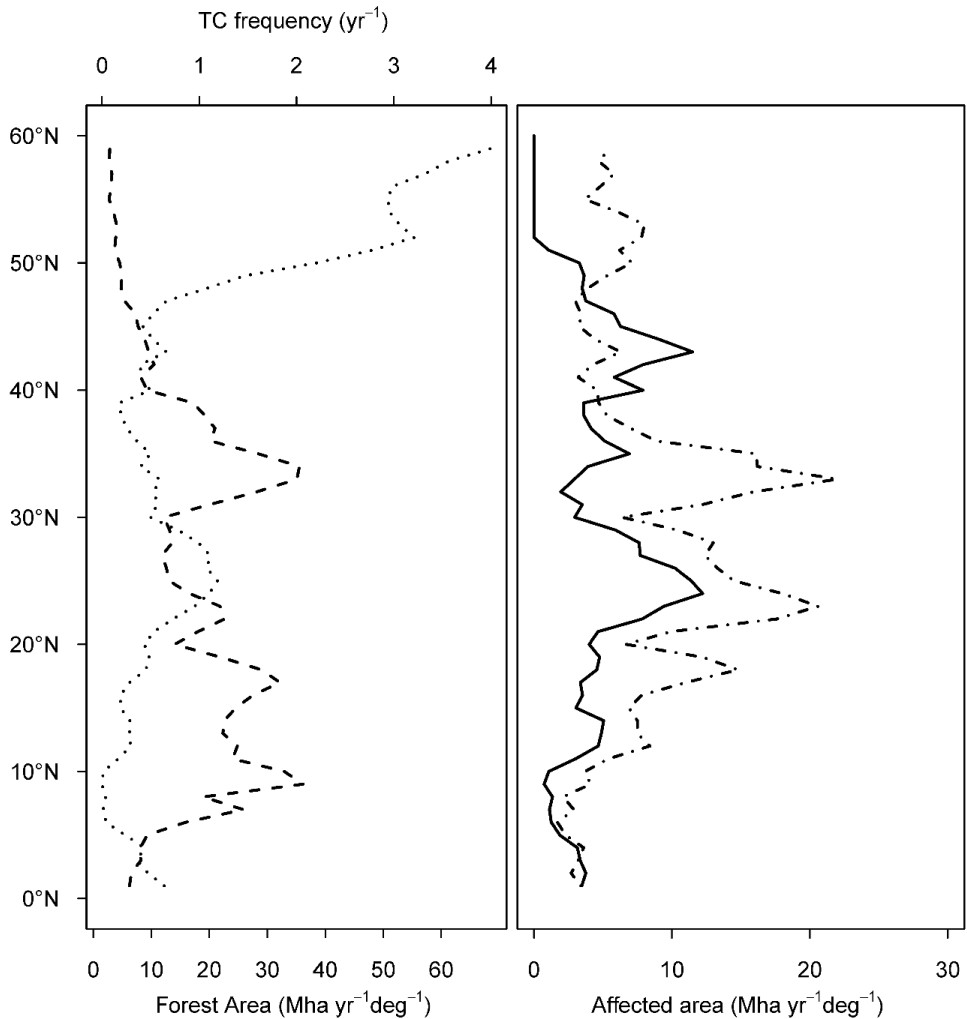


**Figure A1.** Contribution of return frequency and forest cover to the affected area: (**A**) zonal average of forest
coverage (dotted line; Mha deg$^{-1}$) and the return frequency (dashed line; yr$^{-1}$) of TC from 0 degrees N to 60 degrees
N averaged over Eastern Asia, as defined in this study; (**B**) Zonal average of the interaction between return
frequency and forest cover, calculated by multiplying the return frequency with the forest cover (dotdash line; Mha
yr$^{-1}$ deg$^{-1}$) and the estimated zonal average of the annual affected forest area (full line; Mha yr$^{-1}$ deg$^{-1}$) for definition
3b (Table A1). Correlations between return frequency and affected area (Pearson correlation coefficient = -0.35, p-
value < 0.01, n = 60), forest cover and affected area (Pearson correlation coefficient = 0.089, p-value = 0.5, n = 60)





and frequency x cover and affected area (Pearson correlation coefficient = 0.44, p-value < 0.01, n = 60). The latter
thus correlates best with the zonal variation in the affected area and was therefore shown in subplot B.

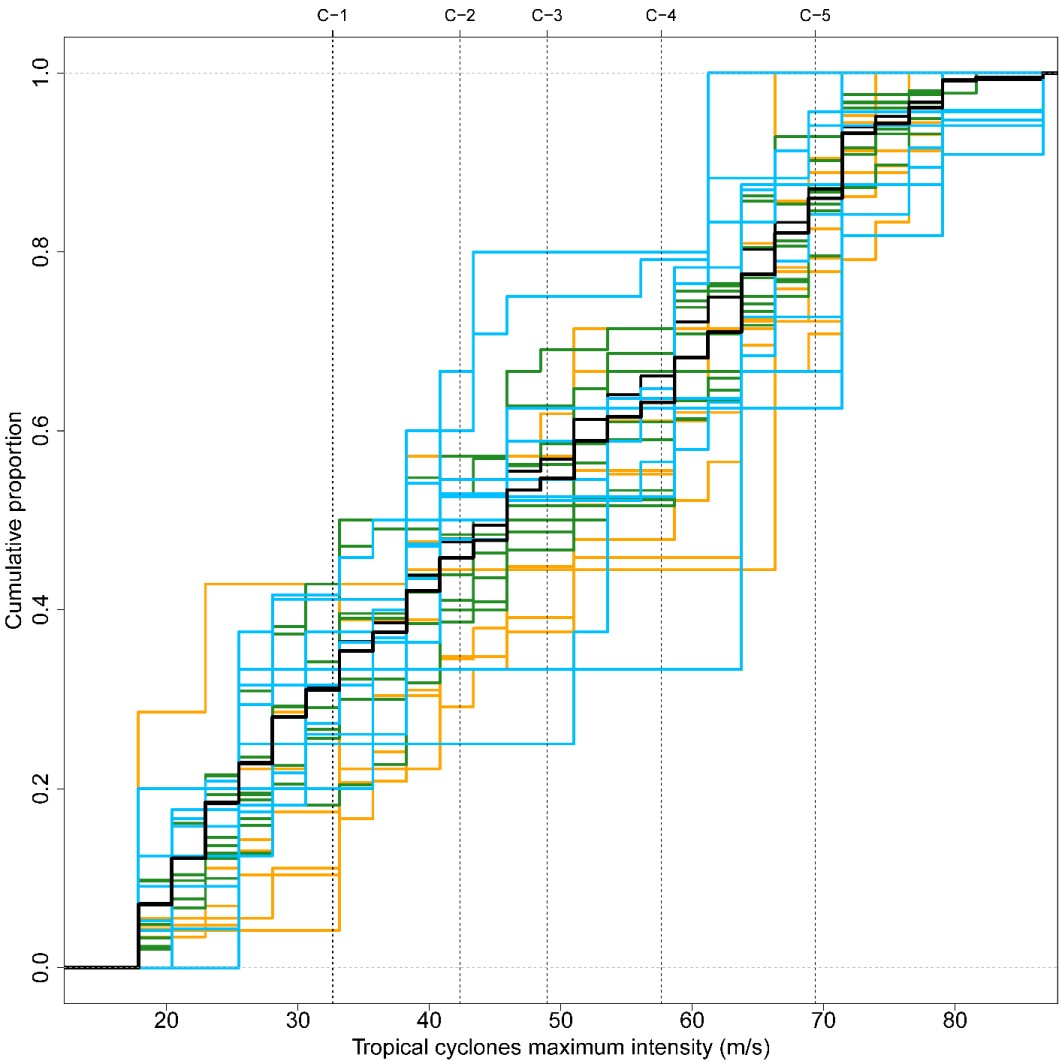


**Figure A2.** Cumulative distribution function of the tropical cyclones as a function of their maximum intensity. The
black solid line shows the distribution of the 580 events that occurred between 1999 to 2018. The grey lines show
the distributions obtained using all nine definitions to calculate the effect sizes: intensity distribution for tropical





cyclones with a negative effect size (orange); intensity distribution for tropical cyclones with a neutral effect size
(blue); and intensity distribution for tropical cyclones with a positive effect size (green).
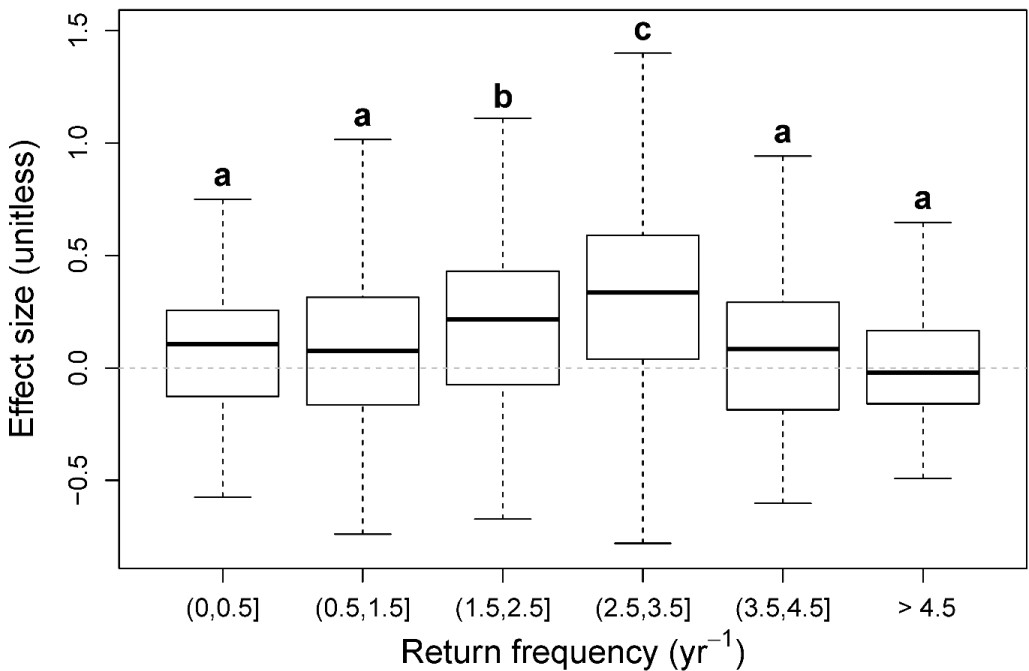


**Figure A3.** Box-wisher plots of the effect size on LAI 60 days following the passage of a tropical cyclone stratified

by the return frequency of the tropical cyclones for the location where the cyclone made landfall. The letters a, b and

c on top of the box whiskers show the different groups identified by a Tukey multiple comparison.



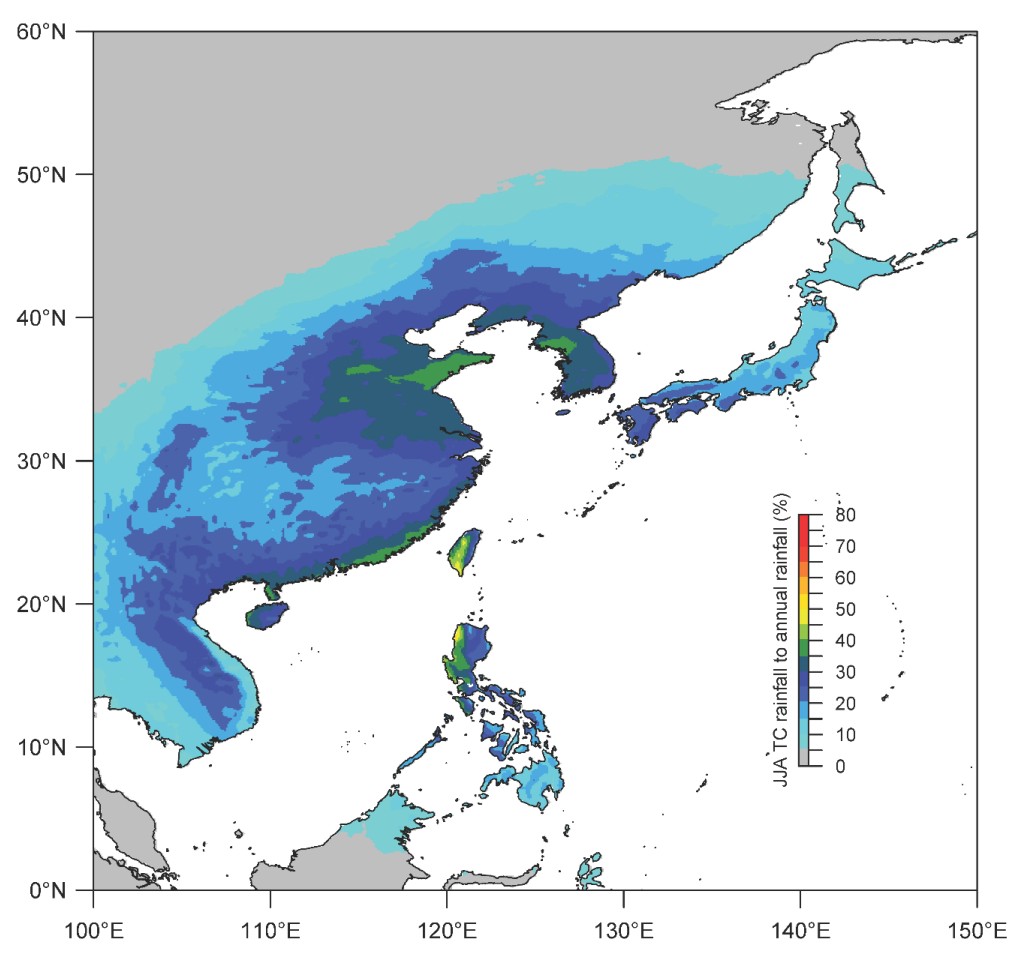


**Figure A4.** Share (%) of the rainfall contributed by tropical cyclones in June, July and August to the total annual
rainfall over Eastern Asia between 1999 to 2018.





**Table A1.** Criteria for distinguishing between the affected and reference areas following the passage of an individual
cyclone and the number of events according to each specific definition. Group 1 groups definitions based on wind
speed, group 2 definitions are based on precipitation and group 3 definitions are based on both wind speed and
precipitation. All three definitions include an estimate of storm path based on a multiple of the reported storm diameter.
Column A denotes the number of events for which data were lacking so that the effect size could not be calculated;
column B denotes the number of events for which all required data were available; column C denotes the subset of B
for which the data passed the quality control (see Quality Control); ES refers to effect size. A total of 580 unique
tropical cyclones were considered in this study.

| Group | Affected area | Reference area | A | B | C | Negative ES | Neutral ES | Positive ES |
|---|---|---|---|---|---|---|---|---|
| 1.a | > 8 ms$^{-1}$ and <2 diameters | < 8 ms$^{-1}$ and <2 diameters | 342 | 238 | 114 | 19 | 62 | 33 |
| 1.b | > 10 ms$^{-1}$ and <3 diameters | < 10 ms$^{-1}$ and <3 diameters | 305 | 275 | 188 | 31 | 113 | 44 |
| 1.c | > 12 ms$^{-1}$ and <4 diameters | < 12 ms$^{-1}$ and <4 diameters | 291 | 289 | 178 | 27 | 105 | 46 |
| 2.a | > 60 mm and <2 diameters | < 60 mm and <2 diameters | 338 | 242 | 117 | 18 | 55 | 44 |
| 2.b | > 80 mm and <3 diameters | < 80 mm and <3 diameters | 315 | 265 | 136 | 10 | 69 | 57 |
| 2.c | > 100 mm and <4 diameters | < 100 mm and <4 diameters | 311 | 269 | 88 | 7 | 36 | 45 |
| 3.a | (> 8 ms$^{-1}$ or > 60 mm) and <2 diameters | (< 8 ms$^{-1}$ or < 60 mm) and < 2 diameters | 352 | 228 | 105 | 21 | 50 | 34 |
| 3.b | (> 10 ms$^{-1}$ or > 80 mm) and <3 diameters | (< 10 ms$^{-1}$ or < 80 mm) and < 3 diameters | 304 | 276 | 196 | 29 | 114 | 53 |
| 3.c | (> 12 ms$^{-1}$ or > 100 mm) and <4 diameters | (< 12 ms$^{-1}$ or < 100 mm) and < 4 diameters | 288 | 292 | 187 | 27 | 110 | 50 |
| Mean | | | 316 | 264 | 145 | 21 | 79 | 45 |
| Std | | | 22 | 22 | 42 | 8 | 31 | 8 |
| Mean (%) | | | 54 | 46 | 25 | 14 | 55 | 31 |
| Std (%) | | | 4 | 4 | 7 | 6 | 21 | 6 |






**Table A2.** Loadings of each characteristic on three axes and collinearity between variables within the same group (See
section "multivariate analysis" for more details). Collinearity was used to build random forests with largely
uncorrelated explanatory variables (**Fig. 2 & 3**). Factor analysis was performed separately for each group. Given the
exploratory nature of this analysis, a factor loading of 0.7 was used as a cut-off and those exceeding that level are
highlighted in bold face. Here, TCC refers to characteristics describing the tropical cyclone itself and PC to the
characteristics of the land and ocean prior to the cyclone.

| Group | Characteristics | FC1 | FC2 | FC3 | Collinearity |
|-------|----------------|-----|-----|-----|--------------|
| TCC | Maximum wind speed during passage over land (m s$^{-1}$) | 0.01 | **0.79** | 0.21 | a |
| | Accumulated rainfall during passage over land (mm) | -0.18 | **-0.83** | 0.08 | b |
| | Latitude of landfall (degrees) | **0.83** | 0.04 | 0.13 | c |
| | Intensity of the tropical cyclone, gusts (m s$^{-1}$) | **0.87** | 0.14 | 0.06 | c |
| | Affected area during passage over land (ha) | 0.15 | 0.09 | **0.97** | d |
| PC | Month of landfall | -0.12 | 0.01 | **0.90** | d |
| | Prior Accumulated rainfall (30 days prior to landfall (mm)) | **0.80** | -0.25 | 0.02 | e |
| | Prior LAI (30 days prior to landfall (m$^2$ m$^{-2}$)) | **0.82** | 0.23 | -0.15 | e |
| | Oceanic Nino index the month of landfall (K) | -0.02 | **0.96** | 0.08 | f |
| | Prior drought state (SPEI, 30 days prior to landfall (mm mm$^{-1}$)) | 0.17 | **-0.71** | 0.32 | g |







**Table A3.** Sets of largely independent variables were used as input in the random forest analysis. Details of the variables are given in the section "multivariate analysis". The justification for the groups is given by the collinearity as reported in Table S2. LAI stands for leaf area index and SPEI stands for Standardized Precipitation Evapotranspiration Index.

| Set | Group with tropical cyclone characteristics (TCC) | Group with land characteristics prior to the cyclone (PC) |
|---|---|---|
| 1 | Maximum wind speed, affected area & latitude | pre-event LAI, oceanic Nino index & month |
| 2 | Accumulated rainfall, affected area & latitude | pre-event LAI, oceanic Nino index & month |
| 3 | Maximum wind speed, cyclone intensity & affected area | pre-event LAI, oceanic Nino index & month |
| 4 | Accumulated rainfall, cyclone intensity & affected area | pre-event LAI, oceanic Nino index & month |
| 5 | Maximum wind speed, affected area & latitude | pre-event SPEI, oceanic Nino index & month |
| 6 | Accumulated rainfall, affected area & latitude | pre-event SPEI, oceanic Nino index & month |
| 7 | Maximum wind speed, cyclone intensity & affected area | pre-event SPEI, oceanic Nino index & month |
| 8 | Accumulated rainfall, cyclone intensity & affected area | pre-event SPEI, oceanic Nino index & month |
| 9 | Maximum wind speed, affected area & latitude | pre-event accumulative rainfall, oceanic Nino index & month |
| 10 | Accumulated rainfall, affected area & latitude | pre-event accumulative rainfall, oceanic Nino index & month |
| 11 | Maximum wind speed, cyclone intensity & affected area | pre-event accumulative rainfall, oceanic Nino index & month |
| 12 | Accumulated rainfall, cyclone intensity & affected area | pre-event accumulative rainfall, oceanic Nino index & month |