# Peer review of "Tropical cyclones facilitate recovery of forest leaf area from dry spells in East Asia"

_Biogeosciences, 2022_

## Author Comment (AC1)

**Referee 1**

We would like to thank both referees for their insightful comments on the original manuscript. Referee 1 and 2 commented on criteria that were used to select/exclude a cyclone from further analyses resulting in a revised set of criteria. These revised criteria will require re-running the entire analysis and remaking all figures and tables. The figures presented in this reply should, therefore, be considered as examples showing how the revised figures could address referee comments but are not final.

Comment 1. It is true that most studies of cyclone (typhoon) disturbance effects focused on major cyclones such that the effects of typhoons on ecosystems are disproportionally derived from studies of these major typhoons. However, I do not think that researchers assume that their studies represent the overall effect of typhoons on ecosystems. I think the studies tried to show that major typhoons could have large impact on ecosystems. With some exceptions, intense typhoons generally caused greater damage. Naturally weak typhoons (e.g., category 1) are unlikely to cause large canopy damage. Thus, I think one piece of information that needs to be added to the manuscript is the proportions of typhoons of different intensity categories for the typhoons passed the selection in this study.

> *One of the reasons why we consider our findings worth to be published in Biogeosciences Letters is that intensity class ranked low as a possible driver in the random forest analysis (Fig. 2). Leaving intensity class out of the random forest analyses decreased the accuracy of the regression trees with less than 10%. Leaving it out might even result in an increase in accuracy (Fig. 2). Nevertheless, we agree with*

*the referee that the intensity categories should be better integrated in the study and*

*we will, therefore, stress this point by:*

*(a) Adding a sentence reporting the share of the different intensity classes (1 to 5) for*

*both the census of 580 cyclones as well as the share for the subsample of cyclones that*

*were further analyzed in this study. This information will be added around L93 of the*

*original submission.*

*(b) Returning to the importance of wind speed in the discussion around L127 of the*

*original submission.*

*(c) Adding the proportion of the different intensity classes in a revised version of Fig.*

*S2. That way the readers can see the distributions for themselves. Following a*

*comment of referee 2, we will show nine subplots to document the differences*

*between the nine definitions for affected areas applied in this study (Table S1). This*

*change also helps to address comment 2 of referee 1. The revised Fig. S2 could look as*

*follows (note that this figure is based on the selection criteria of the original*

*submission. The selection criteria will be revised which will affect all figures and*

*tables):*

[Figure]

*Figure S2. Cumulative distribution functions of tropical cyclones as a function of their maximum intensity for the nine definitions of affected area used in this study. Panel (a) shows wind only for 2 diameters, (b) wind only for 3 diameters, (c) wind only for 4 diameters, (d) rain only for 2 diameters, (e) rain only for 3 diameters, (f) rain only for 4 diameters, (g) wind or rain for 2 diameters, (h) wind or rain for 3 diameters, and (i) wind or rain for 4 diameters as detailed in Table S1. The intensity distribution for tropical cyclones with a negative effect size is shown in orange, for tropical cyclones with a neutral effect size is shown in blue, and for tropical cyclones with a positive effect size in green. The grey solid line shows [to be added in the revised figure] the distribution for the specific definition (n = 145 ± 42 cyclones depending on the definition). The black solid line shows the distribution of the 580*

*events that occurred between 1999 to 2018. Small deviations between the grey and*

*the black line suggest that the sample well represented the 580 cyclones in terms of*

*their intensity class. The thresholds for the intensity classes are based on (Schott et*

*al, 2021). The maximum wind speed of category 1 cyclones is between 32.6 m/s and*

*42.6 m/s, between 4.7 m/s and 49.3 m/s for category 2, between 49.4 m/s and 57.8*

*m/s for category 3, between 57.9 m/s and 69.4 m/s for category 4, and exceeding*

*69.5 m/s for category 5. In East Asia, tropical cyclones of intensity class 3 or higher*

*are called typhoons.*

Comment 2. Related to this, I would also suggest break the analysis by intensity categories of typhoons. If the patterns stay the same (i.e., 30% typhoon did not cause detectable canopy damage) among all categories, that would be a much more interesting finding. If on the other hand, the proportion of no-damage concentrated among weaker typhoons, the results would basically confirmed the findings of previous studies.

> *We agree with the referee and will more prominently feature the intensity categories in the revised manuscript. Proposed revisions addressing this issue are listed in the reply to comment 1 of referee 1.*

Also related to this is the definition of the width of the cyclone track area. I wonder if a more conservative definition is used, would the results stay the same. Because wind velocity decreases with increasing distance from the typhoon eye. A liberal definition is likely to include areas with not strong winds and as such it is not surprising to see limited typhoon impact on forest leaf area.

*As we share this concern with the referee, the original submission introduced nine different definitions for affected area. These definitions varied, as suggested by the referee, the width of the potentially affected area as a function of the cyclone diameter in combination with different climatic thresholds (Table S1). Each definition comprises a combination of at least two out of three criteria, e.g., the diameter of the cyclone, the maximum wind speed at each location during the passage of the cyclone and accumulated precipitation at each location during the passage of the cyclone. This approach was justified in the methods of which a summary follows "Being located within the track of a specific cyclone is essential but not sufficient for damage to occur. Within a storm track, only forested pixels that experienced high wind speeds or high precipitation were counted as in the potentially affected area. Forest pixels that were located within the storm track but did not experience high wind speeds or high precipitation were counted as in the reference area. Each forested pixel within each individual storm track was classified as either affected area or reference area based on these nine definitions. To better account for the uncertainties arising from this approach, the threshold values for wind speed and precipitation were also increased as the track diameter increased (Table S1). For a narrow storm track it is reasonable to assume that there would be damage in all pixels except those where wind speed or precipitation did not exceed a relatively low threshold value. For wide storm tracks the opposite applies; it is reasonable to assume that few of the pixels are damaged except those where wind speed or precipitation exceeded relatively high threshold values".*

*More details on the classification of forest pixels into cyclone-specific reference and affected areas methods can be found in the method section from L193 to L222. Differences in the results coming from differences in the definitions were used throughout the analysis to estimate semantic uncertainties. Figures 1, 2, 3, S2, and S3*

*show this semantic uncertainty. The captions of Figures 4 and S1 mention they shows*

*the result for only definition 3b. Fig S4 does not depend on the nine definitions for*

*affected area (Table S1). It is, therefore, unclear how this comment could be addressed*

*beyond what is already included in the manuscript.*

It is important to note that in situ wind speed experienced by the forests could be very

different from that of the global dataset.

*We agree and this potential mismatch is one of the reasons why we did not base the*

*study on a single definition but used nine definitions instead (Table S1). Likewise, the*

*use of a storm-specific reference and affected area, as well as the many repetitions*

*(145 ± 42) are expected to contribute to robust results.*

Comment 3. The use of images two months following typhoon disturbance bothers me. In

tropical and subtropical region, plant growth could be very quick so that leaf area could

increase substantially in two months, with and without typhoon disturbance. Even for late

typhoons the phenological change could be substantial because most of the affected areas

are in the subtropics with long growing season. Thus, I am concerned that the seemly

positive effect of typhoon on forest leaf area could be an artifact of the long duration

between typhoon passage and image acquisition.

*Exactly this concern convinced us to report an effect size metric rather than an*

*absolute change in LAI. The approach compares, for each individual cyclone, the*

*change in LAI in the affected area against the change in LAI in a cyclone-specific*

*nearby reference area (see equation 1 of the original submission). LAI changes due to*

*leaf phenology are thus accounted for in the effect size. The use of effect sizes makes it*

*very unlikely that positive effect sizes are systematically caused by leaf senescence.*

*The justification of using effect sizes is mentioned in the original submission between*

*L79 and L83 and more details on this approach are given in the method section*

*between L265 and L291.*

*A long time window is more likely to result in neutral effect sizes than in positive or*

*negative effect sizes. Long time windows increase the probability that the LAI*

*recovered from its disturbance which would be reflected in a neutral effect size.*

*Likewise, a long time window increases the changes that the LAI observation following*

*the disturbance occurs after senescence which would also result in a neutral effect*

*size.*

*We, nevertheless, agree with the referee that a two-month period is long and if the LAI*

*product would have allowed to study shorter time windows we would have analyzed*

*the change in LAI 10, 20, 30, 40, 50 and 60 days following the passage of a tropical*

*cyclone. The 60-day period is dictated by the limitation of remote sensed LAI product.*

*In this study, we choose to use the LAI product derived both from high resolution SPOT*

*and PROBV satellite sensor with a very high spatial resolution (1km) and 10 days*

*temporal resolution. As any remote sensing product, this products contains*

*observational gaps which are filled with a local spatial and temporal mean LAI. As this*

*study aims to analyze local spatial and temporal changes in LAI gap filled data could*

*not be used. Using a 60 day window resulted in a reasonable amount of LAI*

*observations for pixels in the reference and the affected area both before and after the*

*passage of a tropical cyclone. The 60-day window is thus more determined by data-*

*availability than disturbance related issues. The justification of the 60-day window will*

*be included in the revised manuscript around L286 of the original submission.*

Comment 4. The most interesting finding of this study is the positive effect of typhoon on leaf area which was attributed to increased water availability. I have several concerns on this finding.

First, as described above most of the increase in leaf area could be from weak typhoons. In this case, it is not surprising because weak typhoons are not expect to have major impact on forests. This has been reported before.

*The random forest analysis suggests that intensity category contributes little to explaining the sign in LAI change. The revisions suggested to address comment 1 by referee 1 will also address this concern. Fig S2 shows that irrespective of which of the nine definitions of the affected and reference area are used, class 3, 4 and 5 typhoons may also result in an increase in LAI 60-day following the passage of the typhoon. The weak relationship between intensity category and the change in LAI is indeed one of the more interesting findings of this study. This will be stressed in the revised manuscript around L126 of the original manuscript.*

Second, also as described above the use of a liberal definition of typhoon track width could also lead to this positive effect because the wind velocity is naturally low in parts of the affected area.

*This could indeed have been a concern if only diameter was used as a criterion but all of our nine definitions for reference and affected areas make use of at least two criteria to overcome this concern. Agreement across the nine definitions suggests that although the data are far from perfect and each individual definition comes with arbitrary thresholds, the results themselves are not very sensitive to these thresholds.*

Third, the two months interval between typhoon passage and image acquisition described above could also lead to the positive effect. A combination of weak intensity, liberal definition of track width and long duration between typhoon passage and image acquisition makes the claim of positive effect of typhoon on forest leaf area problematic. I am not saying that the finding is not true but the above possibilities must be excluded before such a conclusion can be made with confidence. I would also like to see the changes in leaf area in the reference areas during the same period. If leaf area also increased at the similar magnitude, then attributing the effect to typhoons needs more explanation.

*Given the 60-day time window, the LAI is expected to change in the reference area especially if this 60-day window includes the start or the end of the growing season. The effect size needs the change in the reference area to evaluate whether the change in LAI in the affected area is faster, similar or slower. The way the effect size is calculated thus accounts for phenological changes in LAI. If the reference area would not be used in the calculation of the effect size, the change in LAI over the affected area would mostly represent leaf phenology and would thus be unsuitable to address the question at hand. This will be stressed in the method section where the calculation of the effect size is detailed (L265-297).*

*The concern about a liberal definition has been addressed by using an ensemble of nine definitions throughout the study. See reply to comment 2, referee 1.*

*As positive effects are not limited to the cyclones from a low intensity class (Fig. S2), and intensity class has little explanatory power (Fig. 2) a systematic bias is highly unlikely. Given the 60-day time window, our method is more likely to be biased towards detecting no changes in LAI than to detecting positive or negative changes in*

*LAI. This bias expectation will also be added into the method section where the calculation of the effect size is detailed (L265-297).*

Comment 5. The justification of the area LAI difference by using 0.5 which is not consisting with the same shared of the area mean LAI value.

*Thank you for raising the issue. Fig. 6 displayed in the report discussing the quality of the LAI product (Jorge, 2018) used in this study indeed suggests that making the uncertainty proportional to its absolute value is justified. Given that a proportional uncertainty will be stricter than the previously used fixed uncertainty, all analyses presented in the manuscript will have to be rerun. We will adjust this threshold to be 15% difference of the mean LAI value between reference and affected area and rerun all analyses. The section describing the quality control will be adjusted accordingly (around L300-317 in the original submission).*

Comment 6. The figures are mainly about the statistical results. I do not see any results on the actual leaf area and it changes. Thus, the paper is more statistical than ecological/biological.

*This study makes indeed use of statistical approaches to obtain new insights in disturbance ecology. To address this concern of the referee we propose to add an additional figure that ties atmospheric science to disturbance ecology as already described but not illustrated on L149-159 of the original submission. The new Fig. 4 could look as follows (note that this figure is based on the selection criteria of the original submission. The selection criteria will be revised which will affect all figures and tables).*

[Figure]

*Figure 4. Pressure fields (Pa) and changes therein 1 day prior to the passage of a tropical cyclone for cyclones that had a neutral, positive, or negative impact on the leaf area (m2 m-2) of forests. (a) Mean atmospheric pressure and leaf area 1 day prior to the passage of a tropical cyclone that had a neutral impact on forest leaf area. (b) Changes in mean atmospheric pressure and leaf area between cyclones with a neutral and positive effect on leaf area. (c) Changes in mean atmospheric pressure and leaf area between cyclones with a neutral and negative effect on leaf area. Results are shown for affected areas defined as locations within an area extending to three times the cyclone width for which the wind or precipitation exceeded a threshold (Table S1)*

Comment 7. I am not all convinced that less than 1/3 of the typhoons passed the quality control check is representative of the overall typhoons in the region.

*It is correct that depending on the definition used, between 15 and 34 % of the of 580 cyclones could be retained for further analysis. As is the case for most observational studies, there was a trade-off between data quality and number of repetitions. We preferred to use smaller samples sizes in favor of more strict data quality criteria. The smallest sample still contained 88 cyclones (for definition 2c) whereas the largest*

*sample contained 196 cyclones (for definition 3b; Table S1). These are still very*

*reasonable numbers for ecological studies. More importantly, these samples well*

*represent the entire population of the 580 tropical cyclones in the region with regards*

*maximum wind speed (revised Fig. S2, shown under comment 1 for referee 1). Details*

*on how the representativeness of the samples in terms of intensity class will be better*

*stressed in the revised manuscript are given in the reply to comment 1 of referee 1.*

**References used in the replies**

Jorge, S.-Z. *Copernicus Global Land Operations "Vegetation and Energy"*. PP-51, 2018. Link

Schott, T. *et al.* The Saffir-Simpson Hurricane Wind ScaleSaffir-Simpson. PP-4, 2021. Link

---

## Author Comment (AC2)

**Referee 2**

We would like to thank both referees for their insightful comments on the original manuscript. Referee 1 and 2 commented on criteria that were used to select/exclude a cyclone from further analyses resulting in a revised set of criteria. These revised criteria will require re-running the entire analysis and remaking all figures and tables. The figures presented in this reply should, therefore, be considered as examples showing how the revised figures could address referee comments but are not final.

Comment 1. I do have many issues with how certain studies are cited (see line comments), so I think the attribution of findings needs to be done far more carefully. This is my main criticism of the study, so I hope if there is a revised manuscript, that the attribution of citations will not have so many large mistakes.

> *We will carefully check our citations and come back to this issue when discussion the line comments of referee 2.*

Also, it might be a bit contrived to state that it is surprising that cyclones could benefit LAI by increased rainfall. Cyclones bring rainfall over a larger area than the area where they deliver high wind speeds. But this is not a big issue, as it is good to quantify these things. I would argue that the title is a bit too broad and assertive of the occasional positive precipitation effect on LAI. We should keep in mind that this is an analysis focused on the (satellite estimated) LAI of East Asian forests, but that there are several other important other aspects relating to forest response to cyclones that this study does not address (e.g. tree mortality and damage, branchfall, landslides, floods, etc). In my opinion, and even in light of these results, the current title overstates the importance of precipitation on forest responses to cyclones.

> *We see the point made by the referee and propose to change the title in "Tropical cyclones facilitate recovery of forest leaf area from summer droughts in East Asia". Some of the comments by referee 1 and referee 2 suggested that we partly failed in highlighting the novelty of this study. We hope that this new title that will be supported by a thoroughly revised discussion better stresses the take home message of the study.*

> *Note that tree mortality due to turnover or stem breakage, branchfall, and landslides could be expected to result in direct changes in LAI and are, therefore, largely accounted for in the*

*analysis. Given the time window used in this study, legacy effects that emerged 60-days after the passage of the cyclone are not accounted for.*

Comment 2. Line comments. Mha (mega hectares?) is not easy to interpret as a unit. I suggest the authors convert this to km2.

*Mha is indeed mega hectares. We will change this units in the text as well as in Figs 1 and S1.*

A lot of unnecessary acronyms are used, which make the MS more difficult to read. Given the looser length requirements of Biogeosciences, I suggest using as few acronyms as possible.

*Agree with the suggestion. The following acronyms will be removed from a revised manuscript: SPEI (standardized precipitation and evapotranspiration index), LAI (leaf area index), ES (effect size), JTWC (Joint Typhoon Warning Center).*

Some of the sentences are overly long. Reducing these run-on sentences would help.

*Sentence length will be checked while revising the manuscript.*

A figure, or alteration to one of the existing figures, would be useful for the reader to understand where forests currently exist in the region.

*This information can already be found in the Fig.1 in which the shaded and color pixels show the forest cover. We will better describe this in the caption of Fig. 1.*

It is unclear how much of a buffer was applied to the central track of each storm for selecting which pixel locations were affected by cyclones.

*The search area applied to each cyclone was 2, 3 or 4 times the reported diameter. The diameter information for each cyclone was taken from the JWTC database. We will clarify issue in the main text around L46 of the original submission. The approach has been described in detail in the method section of the original submission (L193-222).*

Minor methodological question: How were pixel locations dealt with that received multiple cyclones within the same year?

*Locations that received multiple cyclones are dealt with in the same way as location that received only one cyclone. Given the approach used in this study, the time frame should not be one year but should be shortened to 60-days. In the revised manuscript we will try to quantify how many pixels (expressed as the percentage to of the total forest area affected by tropical cyclones are hit multiple times within two months. The two months period is justified by the 60-day window used in this study. Although the effect size should not be strongly affected by the occurrence of more than one cyclone within 60 days (due to the use of a cyclone-specific reference area), the change in LAI might be attributed to the wrong cyclone. The current analysis would benefit from a low share of pixels that received multiple cyclones within two months.*

Kudos to the authors for adhering to policies regarding open data and reproducible code. One comment is that the git repository for the code linked on Zenodo is exceptionally large at nearly one GB. Perhaps posting another git repo of the final code (with no commit history to reduce size) would be useful. I could be wrong.

*We will look into this and will try to reduce the size of the files that accompany a revised manuscript.*

Comment 3. Figure 1: This is a nice figure but I have some suggestions that I think will increase its interpretability for the reader. I strongly suggest not to use decimal degrees in the denominator, given the actual area will vary with latitude. I suggest presenting the Affected Area as a fraction of the total area per year.

*This comment made us realize that the unit can indeed be misinterpreted. We aggregated the affected area for 1 degree latitudinal bands. $deg^{-1}$ can and will be dropped from the units.*

I suggest selecting a color-blind friendly color palette for panel a, and a legend to indicate areas where forest is not the dominant land cover.

*We noted that Biogeosciences provides a link to a simulator that shows to those who are not color blind how the figure will look like for color-blind readers. A revised version of Fig 1, 3, 4, S2 and S4 will make use of a color-blind friendly color palette. We will improve the caption of Fig 1 to clarify that blank land pixels have a low forest cover.*

A legend for the different lines would aid interpretation, in addition to a slightly more detailed or paraphrased explanation in the legend. Maybe rename the groups to something more informative (wind, precipitation, wind and precipitation) than groups a, b, c.

*We will add a legend to make it easier to link the different lines to the different definitions. An example of    a "paraphrased explanation" can be found in the revised caption for Fig S2 shown in the reply to comment 1 of referee 1.*

Comment 3. Figure 2: This figure is useful, but I have some suggestions: I suggest adding a legend for the surface and cyclone characteristics.

*We will revise this figure by adding a legend. We will replace the ENSO index by the Japan-Taiwan Pacific index in random forest analysis. This substitution will help linking the results (Figs 2 and 4) to a revised discussion (in line with the new title). Figure 4 is a newly added figure and can be seen in the reply to comment 6 of referee 1.*

Comment 4. Figure 3: Any reason that SPEI is not used in the random forest analysis, but is used in Figure 3?

*SPEI is used both in the random forest and the regression tree. Following internal revisions, we forgot to update the label "prior drought state" in Figure 2. A more consistent label should have read "prior SPEI". We will revise Figure 2 accordingly.*

suggest: "Affect area" -> "Affected area"

*Thanks for spotting. This will be corrected.*

I would have thought the boxplot of the decrease in accuracy always be a positive number?

*A negative importance means that removing a given feature from the model actually improves the performance which is possible because of the use of permutations. If a variable was hardly predictive of the outcome, but still selected for some of the splits, randomly permuting the values of that variable may send some observations down a path in the tree which happens to yield a more accurate predicted value, than the path and predicted value that would have been*

*obtained with the original ordering of the variable. We will clarify this issue around L127 of the*
*original submission.*

It is a bit odd that wind speed (or some other wind metric) is not included here.

*Note that following comments by referee 1 and 2 on the selection criteria all analysis will have*
*be re-run. Given revised selection criteria, wind speed may become more (or less) prominent in*
*the revised manuscript.*

*In the original submission we selected the top 6 drivers (based on the random forest; Fig 2) to*
*build the regression tree (Fig 3). Wind speed ranks 7th in the random forest and was therefore*
*not included in regression tree. Likewise we decided, for clarity reasons, to show only four levels*
*in the regression tree. These are arbitrary choices but no matter how we changed these*
*choices, precipitation comes in first and dominates the regression tree. Which we think is an*
*interesting finding as it seemed that both referees were expecting wind speed (this comment*
*referee 1) or intensity category (referee1 several comments) to be among the most important*
*drivers to explain changes in forest leaf area. Although the effect on leaf area of the*
*precipitation brought by cyclones is easy to grasp (comment 1, referee 2), the frequency of this*
*process is surprising and can be explained by the pressure field (New Fig 4, see reply to*
*comment 6 of referee 1). This pressure field is responsible for summer droughts being ended by*
*tropical cyclones.*

I suggest also briefly describing how this decision tree was derived and selected in the figure caption
text.

*The revised caption will read "Figure 3.    Environmental drivers contributing to an increase of*
*leaf area index following the passage of a tropical cyclone. The fractions of a negative, neutral*
*and positive effect size are listed for each box in respectively orange, blue, and green. The*
*number of events is listed as the percentage of the total number of events in the random tree*
*(n=1309). To reduce the collinearity of the input variables, only the six variables with the*
*highest accuracy (Fig. 2) were used to create a four-layer decision tree."*

The numbers in yellow are not going to be very visible if/when this is formatted.

*We will change the color scheme of Fig 3 (see also comment 3 of referee 2) and ensure that it remains consistent with the colors used in Fig. S2.*

I know this is sort of the single best decision tree from the ensemble, but perhaps it would be good to report something like an R2 value?

*According to our understanding of the R-package used, R2 cannot be easily calculated. As an alternative, the performance of the decision tree could be evaluated by splitting the data in a training and an evaluation set. We will look into this request when preparing a revised manuscript.*

Comment 5. Table A1: I suggest spelling out Effect Size, instead of the ES acronym.

*We will do so in the revised manuscript.*

Comment 6. Figure A1: Copying my comment from Figure 1 -> I strongly suggest not to use decimal degrees in the denominator, given the actual area will vary with latitude. I suggest presenting Forest Area as a fraction of the total area, and presenting Affected area in km^2 yr^-1 km^2 (or just a fraction per year).

*This comment made us realize that the unit can indeed be misinterpreted. We aggregated the affected area for 1 degree latitudinal bands. deg-1 can and will be removed from the units.*

Please spell out 'TC' and add a legend corresponding to the different line types.

*We will do so in the revised Fig A1.*

Comment 7. Figure A2: This figure is quite complicated and I am struggling to interpret it. I suggest using a facet of different panels for each different definition. A legend would also help. Also please remind the reader what C-1 through C-5 are.

*C1-C5 show the different intensity categories according to (Schott et al., 2021). We will follow the suggestion of referee 2 to show this result for each definition and to add a legend. An example of how a revised Fig A2 could look like is presented in the reply to comment 1 of referee 1. Note that the color scheme still needs to be adjusted to the needs of the color-blind*

*and that the numbers underlying this figure have to be revised as a consequence of changing the selection criteria.*

Comment 8. Figure A3: Minor point: doing significance tests on discretized groupings of a continuous variable is generally not advisable from my understanding of best practices in statistics. The authors may wish to consider a regression, or using a nonlinear generalized additive model to show the increase and decline of the effect size with respect to return frequency.

*Agreed. We will replace the test on the discretized groupings by a regression on the entire dataset.*

Comment 9. Figure A4: Nice figure, although the color palette is not suitable for the colorblind. The 0-80% stretch seems to miss the focal part of the distribution of the data. Perhaps rescale the color map from 0-50% to improve the contrasts. TC acronym unnecessary.

*The color palette, the scaling issue of the legend and the acronym will be revised and changed accordingly. The revised figure could look as follows:*

[Figure]

*Figure S4. Share (%) of the rainfall contributed by tropical cyclones in June, July and August to the total annual rainfall over Eastern Asia between 1999 to 2018.*

L34: I suggest stating the name of the product within each citation.

*We will do so in the revised manuscript.*

L74-50: This could be rephrased to be clearer. I suggest using commas to separate clauses.

*This comment is not clear. Something might have gone wrong with the line numbers.*

L133: Would be good to add an average LAI % increase because of the additional rainfall.

*This would indeed be a nice result to provide. We will try to extract this number from our analysis and report it in the revised manuscript around L135 of the original submission.*

L150-151: I don't think this text, or this paragraph, attempting to connect summer dry spells to cyclone generation is really necessary.

*We consider this an essential part of the discussion as it explains the required conditions to have an increase in LAI following the passage of a tropical cyclone. It also provides a meteorological relationship between droughts and tropical cyclones which is essential to accept that summer droughts being ended by tropical cyclones are not just rare events but two events caused by the same atmospheric conditions thus making their occurrence highly correlated.*

*We interpreted the discrepancy between our position and this comment by the referee as an indication that we need to further develop this part of the discussion. We consider adding Fig 4 (see our answer to comment 6, referee 1) into the revised manuscript and add additional discussion around L159 of the original submission. Changes in the discussion should ultimately support the new title "Tropical cyclones facilitate recovery of forest leaf area from summer droughts in East Asia".*

L162: This is a bit confusing to me, or at least the wording is around "forest dwarfing". Is small stature of forests being attributed to confer resistance to cyclone damage?

*Small stature of forests is indeed being suggested as a outcome of natural selection in regions with a high return frequency for cyclones. High return frequency should here be regarded in relation to the longevity of an individual tree. This nuance will be added to the revised manuscript.*

L164-165: "The observed frequency of positive vegetation responses to cyclones suggests that the present day vision of cyclones as agents of destruction" - this statement has problems. First, the

reference to the Negrón-Juárez and Nelson studies is incorrect. These studies did not focus on cyclones, but on Amazonian downbursts (sometimes coming from squall lines), which is a very different meteorological process.

*Thanks for spotting. We will remove the wrongly cited studies from the revised manuscript.*

Second, the following are a couple papers quantifying the negative impacts of cyclones (and hurricanes) on forest biomass or mortality, which are potentially important counterpoints to the assertion that cyclones may be providing a forest benefit.
(Negrón-Juárez et al., 2014 Remote sensing ; https://www.mdpi.com/2072-4292/6/6/5633)
(Negrón-Juárez et al., 2014 Remote Sensing of Environment;
https://doi.org/10.1016/j.rse.2013.09.028)
(Negrón-Juárez et al., 2010 JGR Biogeosciences; https://doi.org/10.1029/2009JG001221)

*Thanks for suggesting these references we will consider citing them in the revised manuscript.*

Otherwise there is a very large literature of forest disturbance impacts from Central to North American hurricanes. However, I take the authors' point that additional rainfall can (occasionally) result in LAI increases.

*We do not contest that individual tropical cyclones might be damaging especially towards the eye of the cyclone. With this study we want to point to a circularity in much of the disturbance ecology, i.e., by selecting the most damaging events for further study, the community might overlook many events (including class 3, class 4 and class 5 typhoons, see Fig S2) which are not damaging or might even result in a mean benefit for forest LAI. Note that a mean benefit does not exclude the possibility of serious damage close the track of the eye. Given the conditions which are needed to observe an increase in LAI, the correct conclusion is not necessarily that tropical cyclones increase LAI but is, more likely, that tropical cyclones help forest to recover from summer droughts (an increase in LAI compared to a reference area that experienced the drought but that did not receive the precipitation from the cyclone). We find this to be the case for 31% of the tropical cyclones in the study regions, which we would not label as "occasionally".*

*We considered this comment as a clear indication that the discussion, conclusion, title and abstract needs to be revised to better stress the nuance of our findings, i.e., the wide-spread*

*antagonistic effect that might occur when in East Asia a drought is followed by a tropical*

*cyclone.*

L170: The Stuivenvolt-Allen et al 2021 paper refers to increased fire weather in northwestern North America. Again, given what the sentence says, I think this citation is used incorrectly.

*This citation was chosen deliberately to stress the uncertainty that may come from teleconnection. Sadly, the prefix "tele" was lost during text editing. If this sentence is retained in the revised manuscript we will add it back to restore the integrity of the sentence and the citation. Most likely this thought will be removed from the discussion as it broadens the discussion too much.*

L294-296: I think the citations are used incorrectly in this paragraph. "By design, the latter approach is not capable of identifying neutral or positive impacts of cyclones on leaf area." All but one of these studies have nothing to do with cyclones - so why would they be discussed with respect to cyclone precipitation? The Ozdogan et al., 2014 study is not about cyclones, but windthrows caused by downbursts and tornados. Honkavaara et al 2013 is about detecting forest damage from winter ice storms. The Forzieri et al 2020 paper (of which the second author is a co-author of) is about large-scale windstorms over Europe - again, not cyclones, typhoons, or hurricanes. I argue the authors should be far more careful in their review of the literature and attribution of citations.

*We reread L294-296 in the light of this comment but disagree with the referee. The sentence reads "…in contrast to studies that attribute decreases in leaf area or related satellite-based indices to different disturbance agents including cyclones (Baumann et al., 2014; Honkavaara et al., 2013; Forzieri et al., 2020) including cyclones (Hayashi et al., 2014)". The use of "different disturbance agents" expands our concern from storm damage to other disturbances such as pests, harvest and fires. To justify broadening our concern we cite studies from different disturbance agents. This sentence continues with "including cyclones" which stresses that the previous part of the sentence did not refer to only cyclones.*

*Given that the sentence confused the referee, we propose to move the citations closer to the relevant part of the sentence as follows "…in contrast to studies that attribute decreases in leaf area or related satellite-based indices to different disturbance agents (Baumann et al., 2014; Honkavaara et al., 2013; Forzieri et al., 2020) including cyclones (Hayashi et al., 2014)".*

L304: This seems odd (or perhaps the phrasing is?), the uncertainty almost certainly scales with the magnitude of the LAI estimate. Is 0.18 the domain mean uncertainty over forests? Also what does 0.18 correspond to - a 95% confidence interval?

*Thank you for raising the issue. Referee 1 made a very similar comment (comment 5). Indeed, Fig. 6 displayed in the report discussing the quality of the LAI product (Jorge, 2018) used in this study suggests that making the uncertainty proportional to its absolute value is justified. Given that a proportional uncertainty will be stricter than the previously used fixed uncertainty, all analyses presented in the manuscript will have to be rerun. We will adjust this threshold to be 15% difference of the mean LAI value between reference and affected area and rerun all analyses (hence, the 0.18 will no longer be used). The section describing the quality control will be adjusted accordingly (around L300-317 in the original submission).*

L306: Minor issue: Should it not be 0.5(sqrt(0.18\*\*2 + 0.18\*\*2)) instead of 0.25(sqrt(0.18\*\*2 + 0.18\*\*2)), because it's within a ±0.25 margin of the affected area?

*Thanks for spotting. There was a typo in the manuscript the text should have read $0.25 = (\sqrt{0.18^2 + 0.18^2})$. This criterion and calculation will no longer be used in the revised manuscript.*

L315: This statement is a bit concerning - "Events for which ES < \delta ES were not further analyzed". Filtering the data on account of small effect sizes will certainly bias any subsequent analysis. I think the way this is written could use some clarification.

*The \delta ES is an estimate of the noise present in the LAI data. ES is the signal. If the signal is smaller than the noise, the signal should not be interpreted. Not doing so would mean that we are over interpreting the results. As we would have to decide whether an ES is positive negative or neutral whereas the results tell us that the noise exceeds the signal and that therefore we cannot come to a conclusion.*

*Nevertheless, our estimate of \delta ES was based on several crude assumptions which resulted, in our opinion, in giving too much weight to a rough estimate. Comment 5 by referee 1 and the previous 3 line comments by referee 2 suggested reviewing the selection criteria. We now propose the following revised and simplified selection criteria (the text below will be added to the method section of the revised manuscript).*

*The calculation of the effect size assumes having a similar leaf area index between the area that will become the affected area and the area that will become the reference area after the passage of a cyclone. If the absolute difference in leaf area index between the reference and the affected area was less than 15%, the effect size calculated for this event was included in subsequent analyses. This can be formalized as:*

$$\left| \frac{\overline{LAI}_{bef\ aff}}{\overline{LAI}_{bef\ ref}} -1 \right| < 0.15$$

*Where the 0.15 represents the 15% threshold that was guided by the specifications of the remotely-sensed leaf area product used in this study (Fig 6 in Jorge, 2018). This quality control criterion reflects the idea that prior to the passage of a tropical cyclone, the LAI needs to be similar in what will become the reference and affected area. If not, changes in leaf area following the passage of the cyclone cannot be assigned to its passage.*

*Following the passage of a tropical cyclone, a change in LAI of less than 15% before and after the passage of the cyclone was, in line with the quality control criterion, too small to be considered substantial. Such events were classified as cyclones with a neutral effect size. This classification was formalized as:*

$$\left| \left( \overline{LAI}_{bef} - \overline{LAI}_{aft} \right)_{aff} - \left( \overline{LAI}_{bef} - \overline{LAI}_{aft} \right)_{ref} \right| < 0.15 * \left( \overline{LAI}_{bef} \right)_{ref}$$

*Due to these changes in the selection criteria all analysis will have to be re-run and all figures and tables will have to be updated when preparing a revised manuscript.*

L319-324: Were cyclone characteristics (2 & 3) matched to the corresponding LAI pixel location, or was this an average for the entire trajectory of the cyclone?

*We took the average value along the trajectory. We will clarify this around L325 of the original submission.*

L327: A cautionary note that the precipitation from ERA5 is known to have strong biases in many locations. I don't suggest reanalyzing this, but perhaps a more recent version of GPCP or GPM IMGERv6 would be better for this.

*We considered using the GPCP product but its spatial resolution was considered too coarse (2.5 degree x 2.5 degree) for this study. Following up on this comment we will compare the GPCP to the ERA5-Land data over the study domain and if informative, add the results to the SI of the revised manuscript.*

L341: This is the citation for the R package "psych", not "factor analysis". By all means cite the R package, but again the attribution of the citation is written incorrectly.

*We agree with the referee and will replace this citation by* Kaiser *(1958).*

L351: Please restate what the reference period was in this section.

*We will do so in the revised manuscript.*

**References used in the replies**

Forzieri, G., *et al*.: A spatially explicit database of wind disturbances in European forests over the period 2000–2018, *Earth Syst. Sci. Data*, 12, 257–276, https://doi.org/10.5194/essd-12-257-2020, 2020.

Honkavaara, E., *et al.*: Automatic storm damage detection in forests using high-altitude photogrammetric imagery, *Remote Sens.*, 5, 1405–1424, https://doi.org/10.3390/rs5031405, 2013.

Jorge, S.-Z.: *Copernicus Global Land Operations "Vegetation and Energy"*. PP-51, 2018. Link

Kaiser, H. F.: The varimax criterion for analytic rotation in factor analysis. *Psychometrika*, 23, 187-200. doi:10.1007/BF02289233, 1958. Link

Baumann, M., et al.: Landsat remote sensing of forest windfall disturbance, *Remote Sens. Environ*., 143, 171–179, https://doi.org/10.1016/j.rse.2013.12.020, 2014.

Schott, T., *et al. The Saffir-Simpson Hurricane Wind ScaleSaffir-Simpson*. PP-4, 2021. Link

Hayashi, M., *et al*.: Quantitative assessment of the impact of typhoon disturbance on a Japanese forest using satellite laser altimetry, *Remote Sens. Environ.*, 156, 216–225, https://doi.org/10.1016/j.rse.2014.09.028, 2014.

---

## Author Response (AR1)

**Referee 1**

We would like to thank both referees for their insightful comments on the original manuscript. Referee 1 and 2 commented on criteria that were used to select/exclude a cyclone from further analyses resulting in a revised set of criteria. Revising the criteria required re-running the entire analysis and remaking all figures and tables.

Comment 1. It is true that most studies of cyclone (typhoon) disturbance effects focused on major cyclones such that the effects of typhoons on ecosystems are disproportionally derived from studies of these major typhoons. However, I do not think that researchers assume that their studies represent the overall effect of typhoons on ecosystems. I think the studies tried to show that major typhoons could have large impact on ecosystems. With some exceptions, intense typhoons generally caused greater damage. Naturally weak typhoons (e.g., category 1) are unlikely to cause large canopy damage. Thus, I think one piece of information that needs to be added to the manuscript is the proportions of typhoons of different intensity categories for the typhoons passed the selection in this study.

*One of the reasons why we consider our findings worth to be published in Biogeosciences Letters is that intensity class ranked low as a possible driver in the random forest analysis (Fig. 2). This results can be explained by Fig A2 in which it is shown that negative, neutral and positive effect sizes occur in all intensity classes and that the cumulative distribution of the nine sample that were analysed in this study well represented distribution of the intensity classes of the census of 580 tropical cyclones in the study domain and period. Nevertheless, we agree with the referee that the intensity categories should be better integrated in the study and we, therefore, stressed this point by:*
*(a) Adding a sentence reporting the share of the different intensity classes (1 to 5) for both the census of 580 cyclones as well as the share for the subsample of cyclones that were further analysed in this study. This information was added around **L89-L91.***
*(b) Returning to the importance of wind speed in the discussion around **L128**.*
(c) *Adding the proportion of the different intensity classes in a revised version of Fig. A2. That way the readers can see the distributions for themselves. Following a comment of referee 2, we showed nine subplots to document the differences between the nine definitions for affected areas applied in this study (Table A1). This change also helped to address comment 2 of referee 1.*

Comment 2. Related to this, I would also suggest break the analysis by intensity categories of typhoons. If the patterns stay the same (i.e., 30% typhoon did not cause detectable canopy damage) among all categories, that would be a much more interesting finding. If on the other hand, the proportion of no-damage concentrated among weaker typhoons, the results would basically confirmed the findings of previous studies.

*We agree with the referee and more prominently featured the intensity categories in the revised manuscript. The revisions are listed in the reply to comment 1 of referee 1.*

Also related to this is the definition of the width of the cyclone track area. I wonder if a more conservative definition is used, would the results stay the same. Because wind velocity decreases with increasing distance from the typhoon eye. A liberal definition is likely to include areas with not strong winds and as such it is not surprising to see limited typhoon impact on forest leaf area.

*As we share this concern with the referee, the original submission already introduced nine different definitions for affected area. These definitions vary, as suggested by the referee, the width of the potentially affected area as a function of the cyclone diameter in combination with different climatic thresholds (Table A1). Each definition comprises a combination of at least two out of three criteria, e.g., the diameter of the cyclone, the maximum wind speed at each location during the passage of the cyclone and accumulated precipitation at each location during the passage of the cyclone. This approach is justified in the methods of which a summary follows "Being located within the track of a specific cyclone is essential but not sufficient for damage to occur. Within a storm track, only forested pixels that experienced high wind speeds or high precipitation were counted as in the potentially affected area. Forest pixels that were located within the storm track but did not experience high wind speeds or high precipitation were counted as in the reference area. Each forested pixel within each individual storm track was classified as either affected area or reference area based on these nine definitions. To better account for the uncertainties arising from this approach, the threshold values for wind speed and precipitation were also increased as the track diameter increased (Table A1). For a narrow storm track it is reasonable to assume that there would be damage in all pixels except those where wind speed or precipitation did not exceed a relatively low threshold value. For wide storm tracks the opposite applies; it is reasonable to assume that few of the pixels are damaged except those where wind speed or precipitation exceeded relatively high threshold values".*

*More details on the classification of forest pixels into cyclone-specific reference and affected areas methods can be found in the method section from **L210 to L226**. Differences in the results coming from differences in the definitions were used throughout the analysis to estimate semantic uncertainties. Figures 1, 2, 3, A2, and A3 show this semantic uncertainty. The captions of Figures 4 and A1 mention they show the result for only definition 3a. Fig 4 does not depend on the nine definitions for affected area (Table A1). It is, therefore, unclear how this comment could be addressed beyond what is already included in the manuscript.*

It is important to note that in situ wind speed experienced by the forests could be very different from that of the global dataset.

*We agree and this potential mismatch is one of the reasons why we did not base the study on a single definition but used nine definitions instead (Table A1). Likewise, the use of a storm-specific reference and affected area, as well as the many repetitions (140 ± 41) are expected to contribute to more robust results.*

Comment 3. The use of images two months following typhoon disturbance bothers me. In tropical and subtropical region, plant growth could be very quick so that leaf area could increase substantially in two months, with and without typhoon disturbance. Even for late typhoons the phenological change could be substantial because most of the affected areas are in the subtropics with long growing season. Thus, I am concerned that the seemly positive effect of typhoon on forest leaf area could be an artifact of the long duration between typhoon passage and image acquisition.

*Exactly this concern convinced us to report an effect size metric rather than an absolute change in LAI. The approach compares, for each individual cyclone, the change in LAI in the affected area against the change in LAI in a cyclone-specific nearby reference area (see equation 1). LAI changes due to leaf phenology are thus accounted for in the effect size. The use of effect sizes makes it very unlikely that positive effect sizes are systematically caused by leaf senescence. The justification of using effect sizes is mentioned between **L75 and L81** and more details on this approach are given in the method section between **L270 and L289**. Furthermore, a long time window is more likely to result in neutral effect sizes than in positive or negative effect sizes. Long time windows increase the probability that the LAI recovered from its disturbance which would be reflected in a neutral effect size. Likewise, a long time window increases the changes that the LAI observation following the disturbance occurs after senescence which would also result in a neutral effect size.*

*We, nevertheless, agree with the referee that a two-month period is long and if the LAI product would have allowed to study shorter time windows we would have analysed the change in LAI 10, 20, 30, 40, 50 and 60 days following the passage of a tropical cyclone. The 60-day period is dictated by the limitation of remote sensed LAI product. In this study, we choose to use the LAI product derived both from high resolution SPOT and PROBV satellite sensor with a very high spatial resolution (1km) and 10 days temporal resolution. As any remote sensing product, this product contains observational gaps which are filled with a local spatial and temporal mean LAI. As this study aims to analyse local spatial and temporal changes in LAI, gap filled data could not be used. Using a 60-day window resulted in a reasonable amount of LAI observations for pixels in the reference and the affected area both before and after the passage of a tropical cyclone. The 60-day window is thus more determined by data-availability than disturbance related issues. The justification of the 60-day window is included in the revised manuscript around **L301** and **L307-318**.*

Comment 4. The most interesting finding of this study is the positive effect of typhoon on leaf area which was attributed to increased water availability. I have several concerns on this finding.
First, as described above most of the increase in leaf area could be from weak typhoons. In this case, it is not surprising because weak typhoons are not expect to have major impact on forests. This has been reported before.

*The random forest analysis suggests that intensity category contributes little to explaining the sign in LAI change. The revisions suggested to address comment 1 by referee 1 will also address this concern. Fig A2 shows that irrespective of which of the nine definitions of the affected and reference area are used, class 3, 4 and 5 typhoons may also result in an increase in LAI 60-day following the passage of the typhoon. The weak relationship between intensity category and the change in LAI is indeed one of the more interesting findings of this study. This was stressed in the revised manuscript around **L127**.*

Second, also as described above the use of a liberal definition of typhoon track width could also lead to this positive effect because the wind velocity is naturally low in parts of the affected area.

*This could indeed have been a concern if only diameter was used as a criterion but all of our nine definitions for reference and affected areas make use of at least two criteria to overcome this concern. Agreement across the nine definitions suggests that although the data are far*

*from perfect and each individual definition comes with arbitrary thresholds, the results themselves are not very sensitive to these thresholds.*

Third, the two months interval between typhoon passage and image acquisition described above could also lead to the positive effect. A combination of weak intensity, liberal definition of track width and long duration between typhoon passage and image acquisition makes the claim of positive effect of typhoon on forest leaf area problematic. I am not saying that the finding is not true but the above possibilities must be excluded before such a conclusion can be made with confidence. I would also like to see the changes in leaf area in the reference areas during the same period. If leaf area also increased at the similar magnitude, then attributing the effect to typhoons needs more explanation.

*Given the 60-day time window, the LAI is expected to change in the reference area especially if this 60-day window includes the start or the end of the growing season. The effect size needs the change in the reference area to evaluate whether the change in LAI in the affected area is faster, similar or slower. The way the effect size is calculated thus accounts for phenological changes in LAI. If the reference area would not be used in the calculation of the effect size, the change in LAI over the affected area would mostly represent leaf phenology and would thus be unsuitable to address the question at hand. This has been stressed in the revised method section where the calculation of the effect size is detailed (**L301-305**).*

*The concern about a liberal definition has been addressed by using an ensemble of nine definitions throughout the study. See reply to comment 2, referee 1.*

*As positive effects are not limited to the cyclones from a low intensity class (Fig. A2), and intensity class has little explanatory power (Fig. 2) a systematic bias is highly unlikely. Given the 60-day time window, our method is more likely to be biased towards detecting no changes in LAI than to detecting positive or negative changes in LAI. This bias expectation has been added into the revised method section where the calculation of the effect size is detailed (**L307-L318**).*

Comment 5. The justification of the area LAI difference by using 0.5 which is not consisting with the same shared of the area mean LAI value.

*Thank you for raising the issue. Making the uncertainty proportional to its absolute value seems indeed justified. Given that a proportional uncertainty will be more strict than the*

*previously used fixed uncertainty, all analyses presented in the manuscript had to be rerun. We adjusted this threshold to be 10 % difference of the mean LAI value between reference and affected area and reran all analyses. The section describing the quality control was adjusted accordingly (**L320-337**).*

Comment 6. The figures are mainly about the statistical results. I do not see any results on the actual leaf area and it changes. Thus, the paper is more statistical than ecological/biological.

*This study makes indeed use of statistical approaches to obtain new insights in disturbance ecology. To address this concern of the referee we added an additional figure (Fig 4) that ties atmospheric science to disturbance ecology as already described but not illustrated on L154-158 of the original submission.*

Comment 7. I am not all convinced that less than 1/3 of the typhoons passed the quality control check is representative of the overall typhoons in the region.

*It is correct that depending on the definition used, between 15 and 34 % of the of 580 cyclones could be retained for further analysis. As is the case for most observational studies, there was a trade-off between data quality and number of repetitions. We preferred to use smaller samples sizes in favour of more strict data quality criteria. The smallest sample still contained 86 cyclones (for definition 2c) whereas the largest sample contained 188 cyclones (for definition 3b; Table A1). These are still very reasonable numbers for ecological studies. More importantly, these samples well represent the entire population of the 580 tropical cyclones in the region with regards maximum wind speed (Fig. A2, shown under comment 1 for referee 1). Details on how the representativeness of the samples in terms of intensity class was better stressed in the revised manuscript are given in the reply to comment 1 of referee 1.*

**Referee 2**

We would like to thank both referees for their insightful comments on the original manuscript. Referee 1 and 2 commented on criteria that were used to select/exclude a cyclone from further analyses resulting in a revised set of criteria. Revising the criteria required re-running the entire analysis and remaking all figures and tables.

Comment 1. I do have many issues with how certain studies are cited (see line comments), so I think the attribution of findings needs to be done far more carefully. This is my main criticism of the study, so I hope if there is a revised manuscript, that the attribution of citations will not have so many large mistakes.

*We carefully checked our citations and elaborate on this issue when discussion the line comments of referee 2.*

Also, it might be a bit contrived to state that it is surprising that cyclones could benefit LAI by increased rainfall. Cyclones bring rainfall over a larger area than the area where they deliver high wind speeds. But this is not a big issue, as it is good to quantify these things. I would argue that the title is a bit too broad and assertive of the occasional positive precipitation effect on LAI. We should keep in mind that this is an analysis focused on the (satellite estimated) LAI of East Asian forests, but that there are several other important other aspects relating to forest response to cyclones that this study does not address (e.g. tree mortality and damage, branchfall, landslides, floods, etc). In my opinion, and even in light of these results, the current title overstates the importance of precipitation on forest responses to cyclones.

*We see the point made by the referee and propose to change the title in "Tropical cyclones facilitate recovery of forest leaf area from summer droughts in East Asia". Some of the comments by referee 1 and referee 2 suggested that we partly failed in highlighting the novelty of this study. We hope that this new title that is supported by a thoroughly revised discussion better stresses the take home message of the study.*

*Note that tree mortality due to turnover or stem breakage, branchfall, and landslides could be expected to result in direct changes in LAI and are, therefore, largely accounted for in the analysis. Given the time window used in this study, legacy effects that emerged 60-day after the passage of the cyclone are not accounted for.*

Comment 2. Line comments. Mha (mega hectares?) is not easy to interpret as a unit. I suggest the authors convert this to km2.

*Mha is indeed mega hectares. We changed this unit in $km^2$ in the text as well as in Figs 1 and A1.*

A lot of unnecessary acronyms are used, which make the MS more difficult to read. Given the looser length requirements of Biogeosciences, I suggest using as few acronyms as possible.

*Agree with the suggestion. The following acronyms were removed from the revised manuscript: SPEI (standardized precipitation and evapotranspiration index), LAI (leaf area index), ES (effect size), JTWC (Joint Typhoon Warning Center).*

Some of the sentences are overly long. Reducing these run-on sentences would help.

*Sentence length was checked, and when possible shortened while revising the manuscript.*

A figure, or alteration to one of the existing figures, would be useful for the reader to understand where forests currently exist in the region.

*This information can be found in the Fig.1 in which the shaded and colour pixels show the forest cover. This is now described in the caption of Fig. 1.*

It is unclear how much of a buffer was applied to the central track of each storm for selecting which pixel locations were affected by cyclones.

*The search area applied to each cyclone was 2, 3 or 4 times the reported diameter. The diameter information for each cyclone was taken from the JWTC database. This issue has been clarified in the main text around **L46**. The approach is described in more detail in the method section (**L205-226**).*

Minor methodological question: How were pixel locations dealt with that received multiple cyclones within the same year?

*Locations that received multiple cyclones are dealt with in the same way as location that received only one cyclone. Given the approach used in this study, the time frame should not be one year but should be shortened to 60-day, i.e., how many pixels received multiple cyclones within 60-day. The two months period is justified by the 60-day window used in this study. Although the effect size should not be strongly affected by the occurrence of more than one cyclone within 60 days (due to the use of a cyclone-specific reference area), the change in LAI might be attributed to the wrong cyclone.*

Kudos to the authors for adhering to policies regarding open data and reproducible code. One comment is that the git repository for the code linked on Zenodo is exceptionally large at nearly one GB. Perhaps posting another git repo of the final code (with no commit history to reduce size) would be useful. I could be wrong.

*We reduced the size of the files that accompany a revised manuscript in git repo in the following link https://github.com/ychenatsinca/LAI_STUDY_EA_V1/commits/v1*

Comment 3. Figure 1: This is a nice figure but I have some suggestions that I think will increase its interpretability for the reader. I strongly suggest not to use decimal degrees in the denominator, given the actual area will vary with latitude. I suggest presenting the Affected Area as a fraction of the total area per year.

*This comment made us realize that the unit can indeed be misinterpreted. We aggregated the affected area for 1 degree latitudinal bands. "deg$^{-1}$" can and were dropped from the units.*

I suggest selecting a color-blind friendly color palette for panel a, and a legend to indicate areas where forest is not the dominant land cover.

*We noted that Biogeosciences provides a link to a simulator that shows to those who are not colour blind how the figure will look like for colour-blind readers. According to that tool the revised Figs 1, 3, 4, S2 and S4 make use of a colour-blind friendly colour palette. We improved the caption of Fig 1 to clarify that blank land pixels have a low forest cover.*

A legend for the different lines would aid interpretation, in addition to a slightly more detailed or paraphrased explanation in the legend. Maybe rename the groups to something more informative (wind, precipitation, wind and precipitation) than groups a, b, c.

*We added a legend to make it easier to link the different lines to the different definitions (Fig 1).*

Comment 3. Figure 2: This figure is useful, but I have some suggestions: I suggest adding a legend for the surface and cyclone characteristics.

*We revised this figure by adding a legend in the caption. We replaced the ENSO index by the Pacific Japan index in random forest analysis. This substitution helped linking the results (Figs 2 and 4) to the revised discussion (in line with the new title). Figure 4 is a new figure that was added to the revised manuscript.*

Comment 4. Figure 3: Any reason that SPEI is not used in the random forest analysis, but is used in Figure 3?

*SPEI is used both in the random forest and the regression tree. Following internal revisions we forgot to update the label "prior drought state" in Figure 2. A more consistent label should have read "prior SPEI". Figure 2 was revised accordingly.*

suggest: "Affect area" -> "Affected area"

*Thanks for spotting. This was corrected.*

I would have thought the boxplot of the decrease in accuracy always be a positive number?

*A negative importance means that removing a given feature from the model actually improves the performance which is possible because of the use of permutations. If a variable was hardly predictive of the outcome, but still selected for some of the splits, randomly permuting the values of that variable may send some observations down a path in the tree which happens to yield a more accurate predicted value, than the path and predicted value that would have been obtained with the original ordering of the variable. We clarified this issue around **L124**.*

It is a bit odd that wind speed (or some other wind metric) is not included here.

*Note that following comments by referee 1 and 2 on the selection criteria all analysis had to be re-run. Given the revised selection criteria, wind speed became a bit more prominent in the revised manuscript but it is still has a very low explanatory power. The reason for this low explanatory power is shown in Fig A2 and has been added to the revised manuscript at **L127-129***.

*We selected the top 6 drivers (based on the random forest; Fig 2) to build the regression tree (Fig 3). Likewise we decided, for clarity reasons, to show only four levels in the regression tree.*

*These are arbitrary choices but no matter how we changed these choices, precipitation comes in first and dominates the regression tree. Which we think is an interesting finding as it seemed that both referees were expecting wind speed (this comment referee 1) or intensity category (referee1 several comments) to be among the most important drivers to explain changes in forest leaf area. Although the effect on leaf area of the precipitation brought by cyclones is easy to grasp (comment 1, referee 2), the frequency of this process is surprising and can be explained by the pressure field (New Fig 4, see reply to comment 6 of referee 1). This pressure field is responsible for summer droughts being ended by tropical cyclones.*

I suggest also briefly describing how this decision tree was derived and selected in the figure caption text.

*The revised caption reads "*Environmental drivers contributing to an increase in leaf area in the affected compared to the reference area, following the passage of a tropical cyclone. The fractions of a negative (left), neutral (middle) and positive (right) effect size are shown in each box. The number of events is listed as the percentage of the total number of events in the random tree (n=1262). To reduce the collinearity of the input variables, only the six variables with the highest accuracy (**Fig. 2**) were used to create the four-layer decision tree.*"*

The numbers in yellow are not going to be very visible if/when this is formatted.

*We changed the colour scheme of Fig 3 (see also comment 3 of referee 2) to black and white.*

I know this is sort of the single best decision tree from the ensemble, but perhaps it would be good to report something like an R2 value?

*According to our understanding of the R-package used, R2 cannot be easily calculated.*

Comment 5. Table A1: I suggest spelling out Effect Size, instead of the ES acronym.

*Done in the revised manuscript.*

Comment 6. Figure A1: Copying my comment from Figure 1 -> I strongly suggest not to use decimal degrees in the denominator, given the actual area will vary with latitude. I suggest presenting Forest

Area as a fraction of the total area, and presenting Affected area in km^2 yr^-1 km^2 (or just a fraction per year).

*This comment made us realize that the unit can indeed be misinterpreted. We aggregated the affected area for 1 degree latitudinal bands. $deg^{-1}$ could be and was removed from the units.*

Please spell out 'TC' and add a legend corresponding to the different line types.

*Done.*

Comment 7. Figure A2: This figure is quite complicated and I am struggling to interpret it. I suggest using a facet of different panels for each different definition. A legend would also help. Also please remind the reader what C-1 through C-5 are.

*C1-C5 show the different intensity categories. We followed the suggestion of referee 2 to show this result for each definition and to add a legend.*

Comment 8. Figure A3: Minor point: doing significance tests on discretized groupings of a continuous variable is generally not advisable from my understanding of best practices in statistics. The authors may wish to consider a regression, or using a nonlinear generalized additive model to show the increase and decline of the effect size with respect to return frequency.

*Agreed. We replaced the test on the discretized groupings by a regression on the entire dataset but decided to no longer use it. This figure was removed.*

Comment 9. Figure A4: Nice figure, although the color palette is not suitable for the colorblind. The 0-80% stretch seems to miss the focal part of the distribution of the data. Perhaps rescale the color map from 0-50% to improve the contrasts. TC acronym unnecessary.

*The colour palette, the scaling issue of the legend and the acronym were revised and changed accordingly.*

L34: I suggest stating the name of the product within each citation.

*Done.*

L74-50: This could be rephrased to be clearer. I suggest using commas to separate clauses.

*This comment is not clear. Something might have gone wrong with the line numbers.*

L133: Would be good to add an average LAI % increase because of the additional rainfall.

*We extracted this number from our analysis (Fig. 4) and report it in the revised manuscript around **L134**.*

L150-151: I don't think this text, or this paragraph, attempting to connect summer dry spells to cyclone generation is really necessary.

*We consider this an essential part of the discussion as it explains the required conditions to have an increase in LAI following the passage of a tropical cyclone. It also provides a meteorological relationship between droughts and tropical cyclones which is essential to accept that summer droughts being ended by tropical cyclones are not just rare events but two events caused by the same atmospheric conditions thus making their occurrence highly correlated.*

*We interpreted the discrepancy between our position and this comment by the referee as an indication that we need to further develop this part of the discussion. To this aim, we added Fig 4 (see our answer to comment 6, referee 1) into the revised manuscript and added additional discussion around **L159**. Changes in the discussion support the new title "Tropical cyclones facilitate recovery of forest leaf area from summer droughts in East Asia".*

L162: This is a bit confusing to me, or at least the wording is around "forest dwarfing". Is small stature of forests being attributed to confer resistance to cyclone damage?

*Small stature of forests is indeed being suggested as an outcome of natural selection in regions with a high return frequency for cyclones. High return frequency should here be regarded in relation to the longevity of an individual tree. This was added to the revised manuscript (**L102**).*

L164-165: "The observed frequency of positive vegetation responses to cyclones suggests that the present day vision of cyclones as agents of destruction" - this statement has problems. First, the reference to the Negrón-Juárez and Nelson studies is incorrect. These studies did not focus on

cyclones, but on Amazonian downbursts (sometimes coming from squall lines), which is a very different meteorological process.

*Thanks for spotting. We removed the wrongly cited studies from the revised manuscript.*

Second, the following are a couple papers quantifying the negative impacts of cyclones (and hurricanes) on forest biomass or mortality, which are potentially important counterpoints to the assertion that cyclones may be providing a forest benefit.
(Negrón-Juárez et al., 2014 Remote sensing ; https://www.mdpi.com/2072-4292/6/6/5633)
(Negrón-Juárez et al., 2014 Remote Sensing of Environment;
https://doi.org/10.1016/j.rse.2013.09.028)
(Negrón-Juárez et al., 2010 JGR Biogeosciences; https://doi.org/10.1029/2009JG001221)

*Thanks for suggesting these references we cited them in the revised manuscript.*

Otherwise there is a very large literature of forest disturbance impacts from Central to North American hurricanes. However, I take the authors' point that additional rainfall can (occasionally) result in LAI increases.

*We do not contest that individual tropical cyclones might be damaging especially towards the eye of the cyclone. With this study we want to point to a circularity in much of the disturbance ecology, i.e., by selecting the most damaging events for further study, the community might overlook many events (including class 3, class 4 and class 5 typhoons, see Fig A2) which are not damaging or might even result in a mean benefit for forest LAI. Note that a mean benefit does not exclude the possibility of serious damage close the track of the eye. Given the conditions which are needed to observe an increase in LAI, the correct conclusion is not necessarily that tropical cyclones increase LAI but is, more likely, that tropical cyclones help forest to recover from summer droughts (an increase in LAI compared to a reference area that experienced the drought but that did not receive the precipitation from the cyclone). We find this to be the case for 38 % of the tropical cyclones in the study regions, which we would not label as "occasionally".*

*We considered this comment as a clear indication that the discussion, conclusion, title and abstract had to be revised to better stress the nuance of our findings, i.e., the wide-spread*

*antagonistic effect that might occur when in East Asia a drought is followed by a tropical*

*cyclone.*

L170: The Stuivenvolt-Allen et al 2021 paper refers to increased fire weather in northwestern North America. Again, given what the sentence says, I think this citation is used incorrectly.

*This citation was chosen deliberately to stress the uncertainty that may come from teleconnection. Sadly, the prefix "tele" was lost during text editing. It was decided to remove this thought as it unnecessarily broadened the discussion*

L294-296: I think the citations are used incorrectly in this paragraph. "By design, the latter approach is not capable of identifying neutral or positive impacts of cyclones on leaf area." All but one of these studies have nothing to do with cyclones - so why would they be discussed with respect to cyclone precipitation? The Ozdogan et al., 2014 study is not about cyclones, but windthrows caused by downbursts and tornados. Honkavaara et al 2013 is about detecting forest damage from winter ice storms. The Forzieri et al 2020 paper (of which the second author is a co-author of) is about large-scale windstorms over Europe - again, not cyclones, typhoons, or hurricanes. I argue the authors should be far more careful in their review of the literature and attribution of citations.

*We reread L294-296 of the original manuscript in the light of this comment but disagree with the referee. The sentence reads "…in contrast to studies that attribute decreases in leaf area or related satellite-based indices to different disturbance agents including cyclones (REFs)". The use of "different disturbance agents" expands our concern from storm damage to other disturbances such as pests, harvest and fires. To justify broadening our concern we cite studies from different disturbance agents. This sentence continues with "including cyclones" which stresses that the previous part of the sentence did not refer to only cyclones.*

*Given that the sentence confused the referee, we moved the citations closer to the relevant part of the sentence as follows "…in contrast to studies that attribute decreases in leaf area or related satellite-based indices to different disturbance agents (Ozdogan et al., 2014; Honkavaara et al., 2013; Forzieri et al., 2020) including cyclones (Takao et al., 2014)".*

L304: This seems odd (or perhaps the phrasing is?), the uncertainty almost certainly scales with the magnitude of the LAI estimate. Is 0.18 the domain mean uncertainty over forests? Also what does 0.18 correspond to - a 95% confidence interval?

*Thank you for raising the issue. Referee 1 made a very similar comment (comment 5). Given that a proportional uncertainty is be more strict than the previously used fixed uncertainty, all analyses presented in the manuscript had to be rerun. We adjusted this threshold to be 10 % difference of the mean LAI value between reference and affected area and rerun all analyses (hence, the 0.18 is no longer be used). The section describing the quality control was adjusted accordingly (**L301-337**).*

L306: Minor issue: Should it not be 0.5(sqrt(0.18**2 + 0.18**2)) instead of 0.25(sqrt(0.18**2 + 0.18**2)), because it's within a ±0.25 margin of the affected area?

*Thanks for spotting. There was a typo in the manuscript the text should have read $0.25 = (\sqrt{0.18^2 + 0.18^2})$. This criterion and calculation are no longer be used in the revised manuscript.*

L315: This statement is a bit concerning - "Events for which ES < \delta ES were not further analyzed". Filtering the data on account of small effect sizes will certainly bias any subsequent analysis. I think the way this is written could use some clarification.

*The \delta ES is an estimate of the noise present in the LAI data. ES is the signal. If the signal is smaller than the noise, the signal should not be interpreted. Not doing so would mean that we are over interpreting the results. As we would have to decide whether an ES is positive negative or neutral whereas the results tell us that the noise exceeds the signal and that therefore we cannot come to a conclusion.*

*Nevertheless, our estimate of \delta ES was based on several crude assumptions which resulted, in our opinion, in giving too much weight to a rough estimate. Comment 5 by referee 1 and the previous 3 line comments by referee 2 suggested reviewing the selection criteria. We now revised and simplified selection criteria and added the following description "The calculation of the effect size assumes having a similar leaf area index between the area that will become the affected area and the area that will become the reference area after the passage of a cyclone. If the absolute difference in leaf area index between the reference and the affected area was less than 10 %, the effect size calculated for this event was included in subsequent analyses. This can be formalized as:*

$$|\frac{\overline{LAI}_{bef\,aff}}{\overline{LAI}_{bef\,ref}} -1| < 0.1$$

*Where the 0.1 represents the 10% threshold that was guided by the specifications of the remotely-sensed leaf area product used in this study (Fig 26 in Jorge, 2020). This quality control criterion reflects the idea that prior to the passage of a tropical cyclone, the LAI needs to be similar in what will become the reference and affected area. If not, changes in leaf area following the passage of the cyclone cannot be assigned to its passage.*

*Following the passage of a tropical cyclone, a change in LAI of less than 10 % before and after the passage of the cyclone was, in line with the quality control criterion, too small to be considered substantial. Such events were classified as cyclones with a neutral effect size. This classification was formalized as:*

$$|(\overline{LAI}_{bef} - \overline{LAI}_{aft})_{aff} - (\overline{LAI}_{bef} - \overline{LAI}_{aft})_{ref}| < 0.10 * (\overline{LAI}_{bef})_{ref}$$

*Due to these changes in the selection criteria all analysis had to be re-run and all figures and tables were updated when preparing the revised manuscript.*

L319-324: Were cyclone characteristics (2 & 3) matched to the corresponding LAI pixel location, or was this an average for the entire trajectory of the cyclone?

*We took the average value along the trajectory. This is now clarified around **L325**.*

L327: A cautionary note that the precipitation from ERA5 is known to have strong biases in many locations. I don't suggest reanalyzing this, but perhaps a more recent version of GPCP or GPM IMGERv6 would be better for this.

*We considered using the GPCP product but its spatial and temporal resolutions were considered too coarse for this study.*

L341: This is the citation for the R package "psych", not "factor analysis". By all means cite the R package, but again the attribution of the citation is written incorrectly.

*We agree with the referee and replaced this citation by Grice (2001)*

*Grice, J. W.: Computing and evaluating factor scores., Psychol. Methods, 6, 430–450, https://doi.org/10.1037/1082-989X.6.4.430, 2001*

L351: Please restate what the reference period was in this section.

*Done.*

---

## Referee Report (RR1)

Overall comment:
I thank the authors for thoroughly addressing my comments in detail. Overall, I think the manuscript has been improved, and will be more accessible to a greater number of readers. If I have not commented on specific responses in the following, please interpret this as my agreement with the author's response and or revision.

Specific comments:
* The new title is more appropriate.
* LAI is a standard abbreviation, but thank you for removing the other abbreviations.
* Figure 4 is nice. The legend text is a bit small, although perhaps the copy editor can resize the figure to take up more page space.
* Figure A3 is nice, and quite useful for understanding the rainfall distribution.

Comment 2:
noted

Comment 3:
It's not a big issue, but I still find the display of fractions (negative, neutral, positive), in Figure 3 a bit difficult to interpret.

Comment 7:
* I do not argue for its removal, but I still do not get much value from Figure A2. It is quite complicated and it is a bit difficult to discern the nuanced differences between subpanels. Perhaps others will gain more insight from it than I have.

Comment 9:
Agreed, I think figure 4 helps clarify this point about tropical cyclones aiding LAI recovery from summer drought stress.

L294: Also I thank the authors for clarifying the use of these references in the main text. I strongly agree with the authors' point that starting the assessment from the actual storm tracks is necessary to reduce bias in the assessment. This approach is sorely needed, although not always possible, in the disturbance ecology literature.

---

## Author Response (AR2)

Dear Editor,

We would like to thank referee #1 for the recognition of our efforts in revising the manuscript, as well as referee #2 and the editor for their valuable comments. Below we summarized our responses in a point-by-point report. At the same time, a new revision of the manuscript was prepared based on the review comments. We are confident that the revision addresses the main issues raised by the referees and hope for a positive outcome of the revision process.

Yours Sincerely,

Yi-Ying Chen and Sebastiaan Luyssaert

**The editor:**

Two reviewers have now looked through your manuscript and both are positive about the revision. R1 is now satisfied that the revision addresses their previous points and recommends publication.

*Thank you for the recognition.*

**Comment 1**

By contrast, while R2 is complimentary of much of the revision, they do raise an important point about the focus on **"drought"** in the title, the narrative and the extent to which this is drawn out in the analysis. In looking through the manuscript I feel R2 makes a fair point. You have two drought metrics: "accumulated rainfall" and "prior SPEI", with your analysis making a link to leaf area. The current text is perhaps quite subtle in its messaging around the role of drought. For example, you argue that cyclones increase soil wetness (SPEI increases) and suggest that due to the timing, the implication is the land surface is dry. I'd suggest that this does not necessarily mean it is in "drought". I think the reviewer is making a reasoned point that you could either add an analysis plot or clarify in the text the extent and intensity of drought.

*This point is well taken and was addressed as follows:*
- *We used Liebig's law of the minimum as the framework for an explicit hypothesis to explain the observed increase in leaf area (L83-87), i.e., "Following Liebig's law of the minimum, the observed increase (or reduced decrease) in leaf area implies that about one third of the cyclones alleviated one or more growth factors that were limiting leaf area prior to the passage of the cyclones. We hypothesis that a dry spell could be the growth limiting factor prior to the cyclone, whereas the precipitation brought by the cyclone could enhance plant growth through mitigating soil dryness".*
- *The statistics, figures and narrative of the manuscript were revised and now test the hypothesis. (L557-562, Table1; L625-630, Fig. A4, and L641-643, Table A2; L527-536, Fig.1; L548-555, Fig.3).*
- *Although the region experiences frequent droughts (when defined as events where SPEI drops below -1.0), we replaced "drought" by "dry spell" to stress that our findings relate to a larger range of plant water stresses (L359-360).*
- *Figure 2 (L539-546) shows the location and frequency of the SPEI index dropping below -1.0 and the spatial correlation with the passage of tropical cyclones.*

**Comment 2**

To extend this point, I was personally a little unclear how to interpret that "accumulated rainfall" was so important in explaining LAI responses in Fig 2, but "prior SPEI" had no role.

*This comment made us decide to revise the statistical approach used in the manuscript. Initially we used a factor analysis to detect collinearity and then filtered out collinearity in the random forest. Because "accumulated rainfall" and "prior SPEI" are correlated, mainly one of them, i.e., accumulated rainfall, was retained. "Prior SPEI" entered the analysis only sporadically when accumulated rainfall was not included. In hindsight the way we dealt with collinearity was too complex, i.e., a factor analysis to feed a random forest with largely uncorrelated variables to create a decision tree. The revised statistical approach uses a factor analysis to propose the main axes explaining ~60% of the variance. These axes are then used in the decision tree. The approach shows the relationship between "accumulated rainfall", "prior SPEI", and effect size on leaf area* and therefore addresses this concern.

Unless I've missed it, I don't see much text afforded this point and given the title of the paper is about drought, I think it is fair to ask for a bit more quantification of the links - either via a figure or via the text.

*The revised manuscript is centered around the links between droughts, cyclones and changes in leaf area. Large parts of the text have been revised, the statistics has been revised in line with the narrative and two new figures (Fig. 1 and 2) were added. Note that to keep the manuscript within the word limit, the previous Fig. 1 (frequency map of cyclones) and Fig. 3 (decision tree) were moved to the Appendix. The previous Fig. 2 (random forest) was deleted following the revised statistical approach.*

**Reviewer #1:**

The revised manuscript has adequately addressed most of my concerns raised in the previous review. By focusing on the recovery of leaf area from summer droughts, many of the previous concerns disappear. However, the current version is imbalanced regarding the two key parts in the title "tropical cyclone" and "summer drought".

**Comment 1**

The description on cyclone disturbance is insufficient but the description on summer drought is in adequate. For a study on the recover from droughts, the prevalence and severity of drought should be clearly described and quantified. Currently, the description in the Introduction is very limited and so is the Discussion. In fact, as a key component, I am surprised to see barely any data in the Results related to droughts.

*The first sentence is confusing us. Two typo's might be the cause of this confusion. We understood this sentence as "The description on cyclone disturbance is sufficient but the description on summer drought is inadequate" and revised the manuscript accordingly:*

- *We used Liebig's law of the minimum as the framework of an explicit hypothesis to explain the observed increase in leaf area (L83-87). The statistics, figures and narrative of the manuscript were revised to test this hypothesis*
- *The new Fig. 2 shows the location and frequency of the SPEI index dropping below -1.0 and the spatial correlation with the passage of tropical cyclones.*
- *The correlation between dry spells and tropical cyclones is further analyzed in Tables 1 and Figs. 1&2c*

**Comment 2**

It is only a small part of Figure 2 (prior accumulated rainfall which is not really drought). In fact, the only result that highlights drought is Figure A3. I do not think this is adequate to show the importance of drought in this study. Thus, I recommend a substantial revision that make droughts in the upfront of the manuscript before I can recommend it for publication by Biogeosciences.

*The relationship with drought has been better developed in the text and figures but we did not follow the advice to make it in the upfront of the manuscript. In our*

*opinion the central theme of the manuscript is the interplay between cyclones, drought, changes in leaf area and their relationship with the atmospheric conditions.*

**Reviewer #2**

Overall comment: I thank the authors for thoroughly addressing my comments in detail. Overall, I think the manuscript has been improved, and will be more accessible to a greater number of readers. If I have not commented on specific responses in the following, please interpret this as my agreement with the author's response and or revision.

**Specific comments:**
The new title is more appropriate.

> *Thank you for the recognition.*

LAI is a standard abbreviation, but thank you for removing the other abbreviations.

> *Thank you for the comment.*

Figure 4 is nice. The legend text is a bit small, although perhaps the copy editor can resize the figure to take up more page space.

> *Texts and the legend in the original Figure 4 have been resized and renumber as the new Figure 2.*

**Figure A3** is nice, and quite useful for understanding the rainfall distribution.

> *The original Figure A3 was no longer needed in the revised manuscript.*

Noted Comment 3: It's not a big issue, but I still find the display of fractions (negative, neutral, positive), in Figure 3 a bit difficult to interpret.

> *A legend for describing the fraction was added to the new decision tree in the Figure A4.*

I do not argue for its removal, but I still do not get much value from **Figure A2**. It is quite complicated and it is a bit difficult to discern the nuanced differences between subpanels. Perhaps others will gain more insight from it than I have.

*The Figure A2 in the original manuscript (A3 in the revised manuscript) aims to show*

*that definition 3a has little bias in terms of cyclone intensity and was therefore used in Figs 2, A1, and 2a&b.*

**Comment 9**: Agreed, I think figure 4 helps clarify this point about tropical cyclones aiding LAI recovery from summer drought stress.

*Thank you for the recognition. The figure was renumbered and became Fig. 3 in the revised manuscript.*

**L294**: Also I thank the authors for clarifying the use of these references in the main text. I strongly agree with the authors' point that starting the assessment from the actual storm tracks is necessary to reduce bias in the assessment. This approach is sorely needed, although not always possible, in the disturbance ecology literature.

*Thank you for the recognition.*